# Planarian stem cells sense the identity of the missing pharynx to launch its targeted regeneration

**Tisha E Bohr, Divya A Shiroor, Carolyn E Adler***

Department of Molecular Medicine, Cornell University College of Veterinary Medicine, Ithaca, United States

**Abstract** In order to regenerate tissues successfully, stem cells must detect injuries and restore missing cell types through largely unknown mechanisms. Planarian flatworms have an extensive stem cell population responsible for regenerating any organ after amputation. Here, we compare planarian stem cell responses to different injuries by either amputation of a single organ, the pharynx, or removal of tissues from other organs by decapitation. We find that planarian stem cells adopt distinct behaviors depending on what tissue is missing to target progenitor and tissue production towards missing tissues. Loss of non-pharyngeal tissues only increases non-pharyngeal progenitors, while pharynx removal selectively triggers division and expansion of pharynx progenitors. By pharmacologically inhibiting either mitosis or activation of the MAP kinase ERK, we identify a narrow window of time during which stem cell division and ERK signaling produces pharynx progenitors necessary for regeneration. These results indicate that planarian stem cells can tailor their output to match the regenerative needs of the animal.

## Introduction

When faced with injury or disease, many animals can repair or even replace damaged tissue. This process of regeneration is observed across animal species, and is often fueled by tissue-resident stem cells (*Bely and Nyberg, 2010*; *Sánchez Alvarado and Tsonis, 2006*; *Tanaka and Reddien, 2011*). In response to injury, stem cells accelerate the production of specific types of differentiated cells to repair damaged tissues. For example, in adult mammals, injuries to the intestine, skin, or lung induce stem cells to increase proliferation rates and alter their differentiation potential (*Buczacki et al., 2013*; *Stabler and Morrisey, 2017*; *Tetteh et al., 2015*; *Tumbar et al., 2004*). These findings suggest that injury can modify the behavior of stem cells to promote repair, but how these changes contribute to tissue regeneration remains unclear.

The freshwater planarian *Schmidtea mediterranea* is an ideal model organism to study the interaction between injury and tissue repair due to their virtually endless ability to regenerate (*Ivankovic et al., 2019*). This ability is driven by an abundant, heterogeneous population of stem cells (*Adler and Sánchez Alvarado, 2015*; *Reddien, 2018*; *Zhu and Pearson, 2016*). Defined by ubiquitous expression of the argonaute transcript *piwi-1* (*Reddien et al., 2005*), the planarian stem cell population consists of pluripotent stem cells capable of reconstituting the entire animal (*Wagner et al., 2011*) and likely organ-specific progenitors (*Figure 1A*; *Scimone et al., 2014a*; *van Wolfswinkel et al., 2014*; *Zeng et al., 2018*). These progenitors express organ-specific transcription factors required for the maintenance and regeneration of planarian organs, including a pharynx, primitive eyes, muscle, intestine, an excretory system and a central nervous system (*Figure 1A*), all enveloped in epithelium (*Roberts-Galbraith and Newmark, 2015*). Expression of specific progenitor markers in *piwi-1*[+] stem cells (*Fincher et al., 2018*; *Plass et al., 2018*; *Scimone et al., 2014a*; *van Wolfswinkel et al., 2014*; *Zeng et al., 2018*) provides an opportunity to

*For correspondence:
cea88@cornell.edu

Competing interests: The authors declare that no competing interests exist.

**eLife digest** Many animals can repair and regrow body parts through a process called regeneration. Tiny flatworms called planaria have some of the greatest regenerative abilities and can regrow their whole bodies from just a small part. They can do this because around a fifth of their body is made of stem cells, which are cells that continuously produce new cells and turn into other cell types through a process called differentiation.

Measuring the gene activity in stem cells from planaria shows that these cells are not all the same. Different groups of stem cells have specific genes turned on which are needed to regrow certain body parts. It is unclear whether all stem cells respond to injuries in the same way, or whether the stem cells that respond are specific to the type of injury. For example, stem cells needed to repair the gut may respond more specifically to gut injuries than to other damage.

Bohr et al. studied how stem cells in planaria respond to different injuries, by comparing an injury to a specific organ to a more serious injury involving several organs. The specific injury was the loss of the pharynx, the feeding organ of the flatworm, while the more serious injury was the loss of the entire head. Within hours of removing the pharynx, stem cells that were poised to develop into pharyngeal cells became much more active than other stem cell types. When the head was removed, however, a wide range of stem cells became active to make the different cell types required to build a head. This suggests that stem cells monitor all body parts and respond rapidly and specifically to injuries.

These findings add to the understanding of regeneration in animal species, which is of great interest for medicine given humans' limited ability to heal. Many of the genetic systems that control regeneration in planaria also exist in humans, but are only active before birth. In the long-term, understanding the key genes in these processes and how they are controlled could allow regeneration to be used to treat human injuries.

link the behavior of organ-specific progenitors with injury by tracking stem cell behavior as organ regeneration initiates.

During homeostasis, planarian stem cells replenish organs by steady proliferation that drives cellular turnover (*Pellettieri and Sánchez Alvarado, 2007*). Within hours of any injury, a general increase in stem cell division occurs, along with vast transcriptional changes (*Baguñà, 1976*; *Gaviño et al., 2013*; *Sandmann et al., 2011*; *Wenemoser et al., 2012*; *Wenemoser and Reddien, 2010*). These changes are only sustained beyond the first day if tissue is removed, in what is referred to as the 'missing tissue response' (*Baguñà, 1976*; *Gaviño et al., 2013*; *Wenemoser and Reddien, 2010*). Activated by injury, the extracellular signal-regulated kinase (ERK) contributes to many of these wound-induced transcriptional changes, in addition to regulating stem cell proliferation, differentiation, and survival (*Owlarn et al., 2017*; *Shiroor et al., 2020*; *Tasaki et al., 2011*). Because these injury-induced changes have predominantly been characterized by analyzing broad stem cell behaviors, how they regulate the transition from homeostasis to regeneration of particular organs are key issues to resolve.

Most planarian organs extend throughout the entire body (*Figure 1A*), and injuries often cause simultaneous damage to multiple organs (*Elliott and Sánchez Alvarado, 2013*). The resulting complex regenerative response has limited our ability to decipher how stem cells respond to damage of particular organs. Unlike most planarian organs, except the eye, the pharynx is anatomically distinct (*Adler and Sánchez Alvarado, 2015*; *Kreshchenko, 2009*). Importantly, it can be completely and selectively removed without perturbing other tissues by brief exposure to sodium azide (*Figure 1B*; *Adler et al., 2014*; *Shiroor et al., 2018*). Because only a single organ is removed, pharynx amputation vastly simplifies the regeneration challenge posed to the animal. Previous work identified the forkhead transcription factor *FoxA* as an essential regulator of pharynx regeneration (*Adler et al., 2014*; *Scimone et al., 2014a*). Under homeostatic conditions, *FoxA* is expressed in the pharynx and a subset of stem cells. Pharynx amputation triggers an increase in $FoxA^+$ stem cells, demonstrating that injury expands the pool of pharynx progenitors. These properties allow us to dissect how stem cells respond to loss of a specific organ and are regulated to restore it.

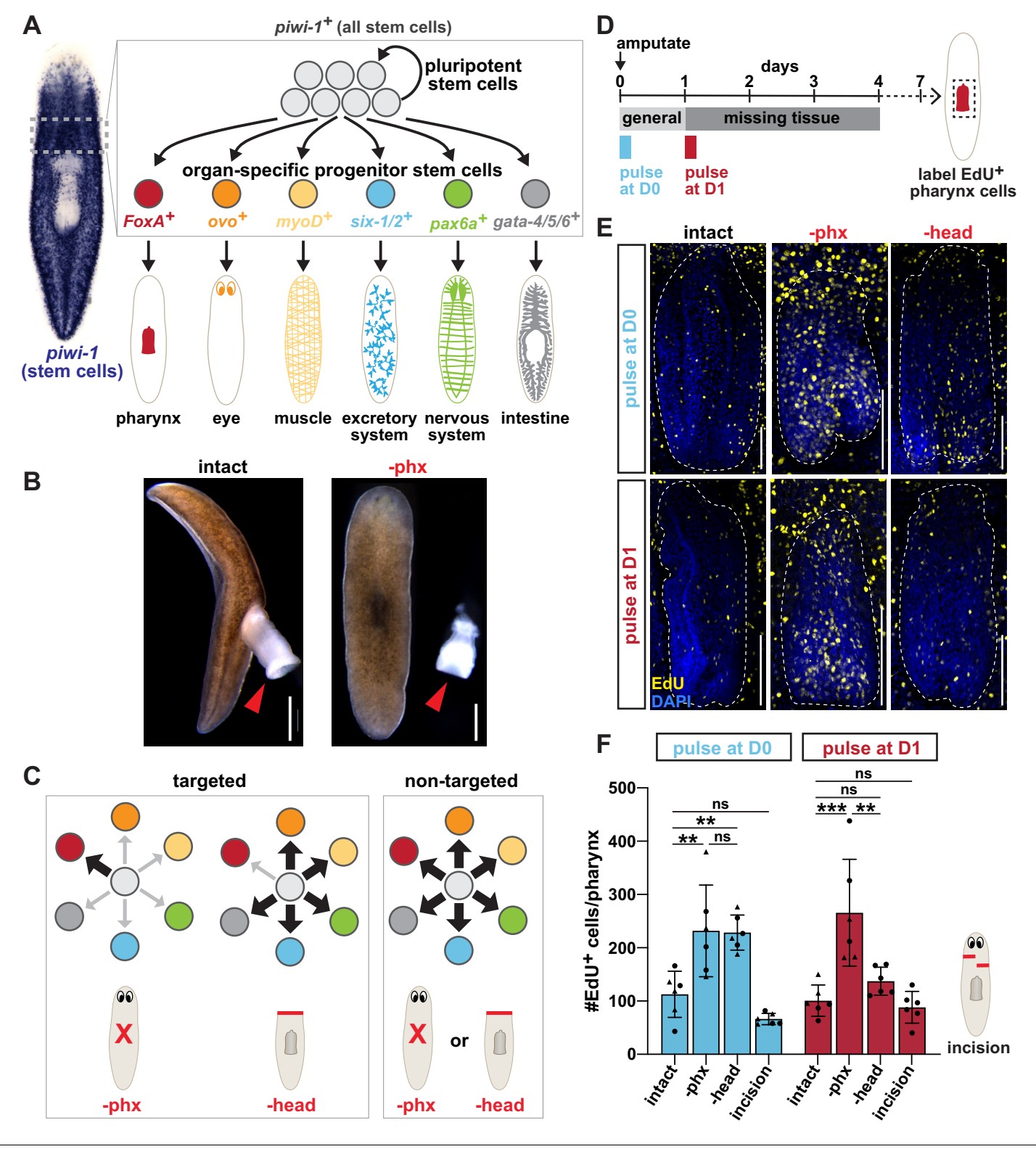

**Figure 1.** Both targeted and non-targeted mechanisms contribute to pharynx regeneration. (**A**) Schematic of planarian stem cell lineage. Left, whole-mount *in situ* hybridization (WISH) for the stem cell marker *piwi-1*. Right, cartoon depiction of the dashed boxed region showing planarian stem cells consisting of pluripotent stem cells and organ-specific progenitors that produce planarian organs. Markers of organ-specific progenitors are indicated. (**B**) Live images of planarians before and after pharynx amputation. Arrows = pharynx; scale bars = 500 μm. (**C**) Models for targeted and non-targeted

*Figure 1 continued on next page*

*Figure 1 continued*

regeneration after different amputations (indicated by red lines). Progenitors are color coded as in A. (D) Schematic of F-ara-EdU delivery relative to amputation for E and F. (E) Confocal images of F-ara-EdU (yellow) in the pharynx (dashed outline) of intact animals, or 7 days after pharynx or head amputation. Dashed box in D = region imaged; DAPI = DNA (blue); scale bar = 100 μm. (F) Number of F-ara-EdU$^+$ cells in the entire pharynx (dashed outline in E). Cartoon represents incision injuries. Graphs represent mean ± SD; symbols = individual animals; shapes distinguish biological replicates and; *, p≤0.05; **, p≤0.01; ***, p≤0.001; one-way ANOVA with Tukey test. Raw data can be found in *Figure 1—source data 1*.

The online version of this article includes the following source data and figure supplement(s) for figure 1:

**Source data 1.** Quantification of F-ara-EdU$^+$ cells in *Figure 1F*.

**Figure supplement 1.** Selective pharynx removal does not increase incorporation of new brain cells.

**Figure supplement 1—source data 1.** Quantification of F-ara-EdU$^+$ cells in *Figure 1—figure supplement 1C*.

The ability of planarians to replace exactly the tissues that have been damaged or removed by injury remains one of the outstanding questions in regeneration (*Mangel et al., 2016*; *Nishimura et al., 2011*). Previous studies have suggested that stem cells selectively increase the output of specific progenitors of depleted organs, implying a targeted mode of regeneration (*Figure 1C*; *Thi-Kim Vu et al., 2015*). However, others have shown that stem cells respond indiscriminately to tissue removal, incorporating new cells into tissues regardless of whether they have been damaged, suggesting a non-targeted mode of regeneration (*LoCascio et al., 2017*). Based on these findings, the authors proposed that stem cells non-selectively increase production of any nearby progenitors, determined by the size and position of a wound, rather than the identity of missing tissues (*Figure 1C*). These two seemingly contradictory models introduce uncertainty into our understanding of the relationship between tissue loss and the stem cell behaviors that ultimately contribute to the regeneration of missing tissues.

By performing an in-depth analysis of specific populations of stem cells in response to different injuries, we show that stem cells can sense the identity of missing tissues. Inflicting various injuries to both the pharynx and body defines distinct contributions of stem cells to regenerated tissue depending on when they divide relative to injury. Planarian stem cells respond to organ loss by selectively increasing expression of organ-specific transcription factors required for subsequent regeneration. Amputation of non-pharyngeal tissues only amplifies non-pharyngeal progenitors, while removal of pharynx tissue selectively increases pharynx progenitors. This increase in pharynx progenitors, and subsequent pharynx regeneration, depends on stem cell division and the MAP kinase ERK, during defined times after tissue loss. Unlike the pharynx, eye regeneration following selective removal is not dependent on stem cell division or ERK signaling, suggesting that different injuries may require distinct regenerative mechanisms. We propose that, in addition to non-targeted and passive modes of regeneration, stem cell behavior can be altered by the loss of specific tissues, selectively channeling their output towards replacement of missing organs.

## Results

### Both targeted and non-targeted mechanisms contribute to pharynx regeneration

Planarian stem cells have distinct responses to injury depending on whether or not tissue has been removed. Any injury induces a proliferative response within hours, while tissue removal causes a sustained response for up to 4 days, leading to localized proliferation and differentiation (*Baguñà, 1976*; *Wenemoser and Reddien, 2010*). Therefore, it has been hypothesized that the initial injury response is a 'general' mechanism for repair, whereas the later 'missing-tissue response' may be tailored to target the replacement of lost tissue. To evaluate the outcome of stem cell proliferation during these specific injury responses, we altered the timing of stem cell labeling relative to different injuries and analyzed the prevalence of labeled cells in mature organs.

We labeled stem cells with the thymidine analogue F-ara-EdU (*Neef and Luedtke, 2011*) for 4 hr either immediately or 1 day after pharynx or head amputation. We then analyzed F-ara-EdU$^+$ cells in the pharynx 7 days after amputation (*Figure 1D*). When F-ara-EdU was applied immediately after amputation (D0), we observed increased F-ara-EdU$^+$ cells in the pharynx following either pharynx or head removal, as compared to intact controls (*Figure 1E,F*). The timing of this pulse, relative to

injury, confirms a previous study showing that amputation stimulates general incorporation of newly generated cells into non-injured tissues (*LoCascio et al., 2017*). However, F-ara-EdU administration 1 day after amputation (D1) resulted in a specific increase in F-ara-EdU$^+$ pharynx cells only after removal of the pharynx, but not the head (*Figure 1E,F*). To determine if increased tissue production requires tissue removal, prior to F-ara-EdU administration we performed incisions anterior to the pharynx, which damaged the body without removing any tissue (*Figure 1F*). However, the number of F-ara-EdU$^+$ cells in the pharynx were comparable to controls, suggesting that tissue removal strongly stimulates production of new tissue, while injury alone does not.

To determine whether other tissues incorporate new stem cells in a time-dependent manner relative to injury, we analyzed the number of newly generated neurons in the brain by combining F-ara-EdU staining with FISH for the neuronal marker *ChAT* (*Figure 1—figure supplement 1A*; *Wagner et al., 2011*). In the newly regenerated brain, head amputation increased F-ara-EdU$^+$ *ChAT$^+$* neurons after both F-ara-EdU pulse conditions, as compared to intact controls. However, the number of F-ara-EdU$^+$ *ChAT$^+$* brain neurons were comparable to controls after either pharynx amputation or incisions, regardless of when the F-ara-EdU pulse was administered (*Figure 1—figure supplement 1B,C*). Because chemical pharynx removal does not increase production of neural tissue (*Figure 1—figure supplement 1B,C*), while its surgical removal does (*LoCascio et al., 2017*), non-targeted regenerative mechanisms may require injury to specific types of tissues, such as body-wall muscle, epithelia, or intestine. In fact, amputation-specific transcriptional changes important for regeneration have recently been identified within these tissues (*Witchley et al., 2013*; *Lander and Petersen, 2016*; *Scimone et al., 2016*; *Benham-Pyle et al., 2020*). These results suggest that while cells generated immediately after tissue removal can be broadly deployed to all surrounding tissues, those generated 1 day later are targeted toward only those that are missing.

## Pharynx tissue loss selectively increases pharynx progenitors

Our data so far indicate that injury channels the output of stem cells towards missing tissues. If this targeted model is true, injuries that do not remove pharynx tissue, like head amputations, should not increase pharynx progenitors (*Figure 1C*). Alternatively, if regeneration is non-targeted, injury should non-selectively increase production of any nearby progenitors (*LoCascio et al., 2017*). If this is the case, head amputation, where pharyngeal tissue is not removed, should also stimulate an increase in pharynx progenitors (*Figure 1C*).

To test the response of organ-specific progenitors (*Figure 1A*) to loss of different tissues, we challenged animals with either head or pharynx amputation and analyzed changes in expression of organ-specific progenitor markers within *piwi-1$^+$* stem cells. First, we labeled pharynx progenitors with double fluorescent *in situ* hybridization (FISH) for *piwi-1* and the pharynx-specific progenitor marker *FoxA*, 3 days after pharynx or head amputation. We then quantified pharynx progenitors in the same region, anterior to the pharynx (*Figure 2A*). As previously reported, we found that pharynx removal caused a significant increase in pharynx progenitors as compared to intact controls (*Adler et al., 2014*; *Scimone et al., 2014a*). By contrast, head amputation did not influence the number of pharynx progenitors, which were similar to intact animals (*Figure 2A,B*). To determine when pharynx progenitors emerge and how long they persist, we quantified the number of pharynx progenitors at various times after pharynx amputation and found that they significantly increased 3 days after amputation (*Figure 2C*). Because injury broadly influences stem cell behavior, we also analyzed the proportion of these pharynx progenitors relative to all other *piwi-1$^+$* stem cells at the same times after pharynx and head amputation and found a similar trend (*Figure 2—figure supplement 1A,B*). These data indicate that pharynx progenitors are selectively produced 3 days after pharynx loss, but not after loss of other tissue types.

To determine which types of injuries stimulate an increase in pharynx progenitors, we inflicted various injuries to or around the pharynx (*Figure 2D*). We then labeled and quantified pharynx progenitors 3 days later. Incisions that damaged the pharynx without removing any tissue failed to stimulate an increase in *piwi-1$^+$FoxA$^+$* stem cells. However, partial removal of the pharynx (~50–80%) caused a significant increase in pharynx progenitors compared to intact controls. We also performed flank resections, which removed tissue from regions adjacent to the pharynx but did not damage the pharynx itself. Despite being previously shown to increase new cells into the uninjured pharynx with BrdU labeling (*LoCascio et al., 2017*), we observed comparable numbers of pharynx progenitors as in intact controls (*Figure 2D*). To determine if the increase in pharynx progenitors was localized, we

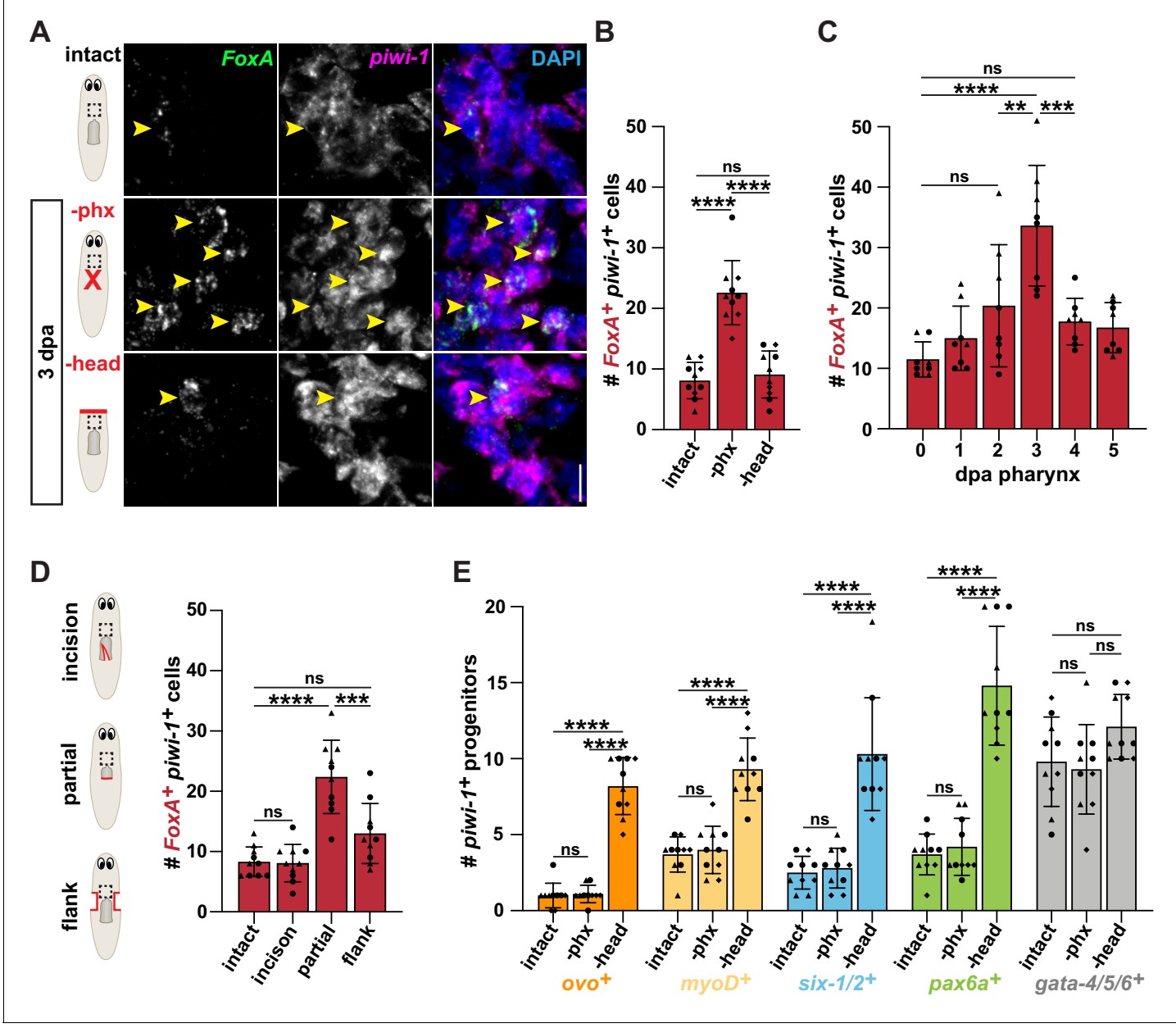

**Figure 2.** Pharynx tissue loss selectively increases pharynx progenitors. (A) Confocal images of double fluorescent *in situ* hybridization (FISH) for *FoxA* (green) and *piwi-1* (magenta) in intact and injured animals, 3 days post-amputation (dpa). Images are partial projections of a portion of the area outlined by dashed boxes. DAPI = DNA (blue); arrows = double-positive cells; scale bar = 10 μm. (B) Number of *FoxA*+ *piwi-1*+ cells in the area outlined by dashed boxes in A. (C) Number of *FoxA*+ *piwi-1*+ cells at indicated times post-pharynx amputation in the area outlined by dashed boxes in A. (D) Number of *FoxA*+ *piwi-1*+ cells in intact and injured animals, 3 days after injury (red lines) in the area outlined by dashed boxes in cartoons. (E) Number of cells double-positive for *piwi-1* and the indicated progenitor marker in intact and injured animals, 3 days after amputation in the area outlined by dashed boxes in A. For graphs, a 6000 μm$^2$ region in the same location of the pre-pharyngeal region was analyzed over 20 z-sections, as represented by dashed boxes in A and D. Graphs represent mean ± SD; symbols = individual animals; shapes distinguish biological replicates and; \*\*, p≤0.01; \*\*\*, p≤0.001; \*\*\*\*, p≤0.0001, one-way ANOVA with Tukey test. Raw data can be found in *Figure 2—source data 1*.

The online version of this article includes the following source data and figure supplement(s) for figure 2:

**Source data 1.** Quantification of *piwi-1*+ cells in *Figure 2B–E*.

**Figure supplement 1.** Amputation increases organ-specific progenitors relative to stem cells and is localized to wounds.

**Figure supplement 1—source data 1.** Quantification of *piwi-1*+ cells in *Figure 2—figure supplement 1A–C*.

**Figure supplement 2.** Pharynx loss does not affect non-pharyngeal progenitors.

**Figure supplement 3.** Non-pharyngeal organ-specific transcription factors are not required for pharynx regeneration.

*Figure 2 continued on next page*

*Figure 2 continued*

**Figure supplement 3—source data 1.** Quantification of feeding behavior in *Figure 2—figure supplement 3A*.

analyzed the same-sized regions in tails, farther from the site of amputation. However, we did not detect an increase in pharynx progenitors in tails after either pharynx or head amputation as compared to intact controls (*Figure 2—figure supplement 1C*). These data suggest that stimulation of pharynx progenitor production is local and requires recognition of lost pharynx tissue, but not necessarily loss of the entire organ.

Based on our finding that pharynx progenitors increase only in response to missing pharynx tissue, we hypothesized that the specific pairing of organ loss and progenitor increase would be true for other organs. Besides the pharynx, the eye is the only other planarian organ that is anatomically restricted and thus can be fully removed without leaving any remaining tissue behind (*Lapan and Reddien, 2012*; *LoCascio et al., 2017*). Expression of the eye-specific transcription factor *ovo* is required for eye regeneration (*Figure 1A*; *Flores et al., 2016*; *Lapan and Reddien, 2012*; *Rouhana et al., 2013*; *Scimone et al., 2011*; *Scimone et al., 2017*; *Scimone et al., 2014a*) and $ovo^+$ $piwi\text{-}1^+$ eye progenitors increase after decapitation (*Lapan and Reddien, 2012*). Conversely, following pharynx amputation, this increase did not occur (*Figure 2E*, *Figure 2—figure supplement 2A*), indicating that pharynx loss does not stimulate the production of eye progenitors.

We also quantified the responses of organ-specific progenitors for muscle ($myoD^+$), intestine ($gata\text{-}4/5/6^+$), the excretory system ($six\text{-}1/2^+$) and the nervous system ($pax6a^+$) (*Figure 1A*; *Flores et al., 2016*; *Scimone et al., 2011*; *Scimone et al., 2017*; *Scimone et al., 2014a*) after either pharynx or head removal. With the exception of intestinal progenitors, all others showed a similar behavior, increasing within 3 days after head removal, but not pharynx removal (*Figure 2E*, *Figure 2—figure supplement 2B–E*). Analysis of the proportion of progenitors relative to all $piwi\text{-}1^+$ stem cells in the same regions yielded comparable outcomes (*Figure 2—figure supplement 1B*). No changes in any of these organ-specific progenitors were observed in tail regions after pharynx or head amputation, indicating that it is a local response (*Figure 2—figure supplement 1C*). Together, these data indicate that planarian stem cells sense the loss of missing tissues to initiate their regeneration through the selective expansion of organ-specific progenitors.

All these organ-specific transcription factors, with the exception of *pax6a*, are required for regeneration of their cognate organ (*Adler and Sánchez Alvarado, 2017*; *Flores et al., 2016*; *Lapan and Reddien, 2012*; *Pineda et al., 2002*; *Scimone et al., 2011*; *Scimone et al., 2017*). Therefore, to test whether or not these transcription factors regulate pharynx regeneration, we knocked them down with RNAi and assayed feeding ability 7 days after pharynx amputation (*Adler et al., 2014*; *Ito et al., 2001*). Unlike *FoxA(RNAi)*, knockdown of other organ-specific progenitor markers did not impact the recovery of feeding ability (*Figure 2—figure supplement 3A*), despite efficient knockdown and manifestation of known phenotypes (*Figure 2—figure supplement 3B*, data not shown). Because *pax6a* is not required for brain regeneration, we performed RNAi of *coe*, a neural progenitor marker that is required for brain regeneration (*Cowles et al., 2013*). However, *coe* knockdown did not affect the recovery of feeding ability (*Figure 2—figure supplement 3A*). Therefore, it is unlikely that any of these other progenitor markers contribute to pharynx regeneration, despite the presence of muscle and neural tissue within the pharynx.

## Pharynx loss selectively induces mitosis of pharynx progenitors

The increase in pharynx progenitors following pharynx amputation suggests that stem cells may divide in response to pharynx loss to selectively amplify pharynx progenitors. Because stem cells are the only dividing cells in planarians (*Morita and Best, 1984*), we can visualize stem cell division with antibody staining for histone H3Ser10 phosphorylation (H3P). Using this mitotic marker, we verified that stem cell division increased near wounds beginning 1 day after either pharynx or head removal (*Figure 3A*; *Adler et al., 2014*; *Baguñà, 1976*). To determine if these dividing stem cells are pharynx progenitors, we combined antibody staining for H3P with FISH for *FoxA*. One day after pharynx removal, we observed higher numbers of $H3P^+$ pharynx progenitors in regions adjacent to wounds as compared to intact animals (*Figure 3B*). To determine how soon after amputation these pharynx progenitors initiate division, and how long it persists, we quantified the coincidence of $FoxA^+$ $H3P^+$

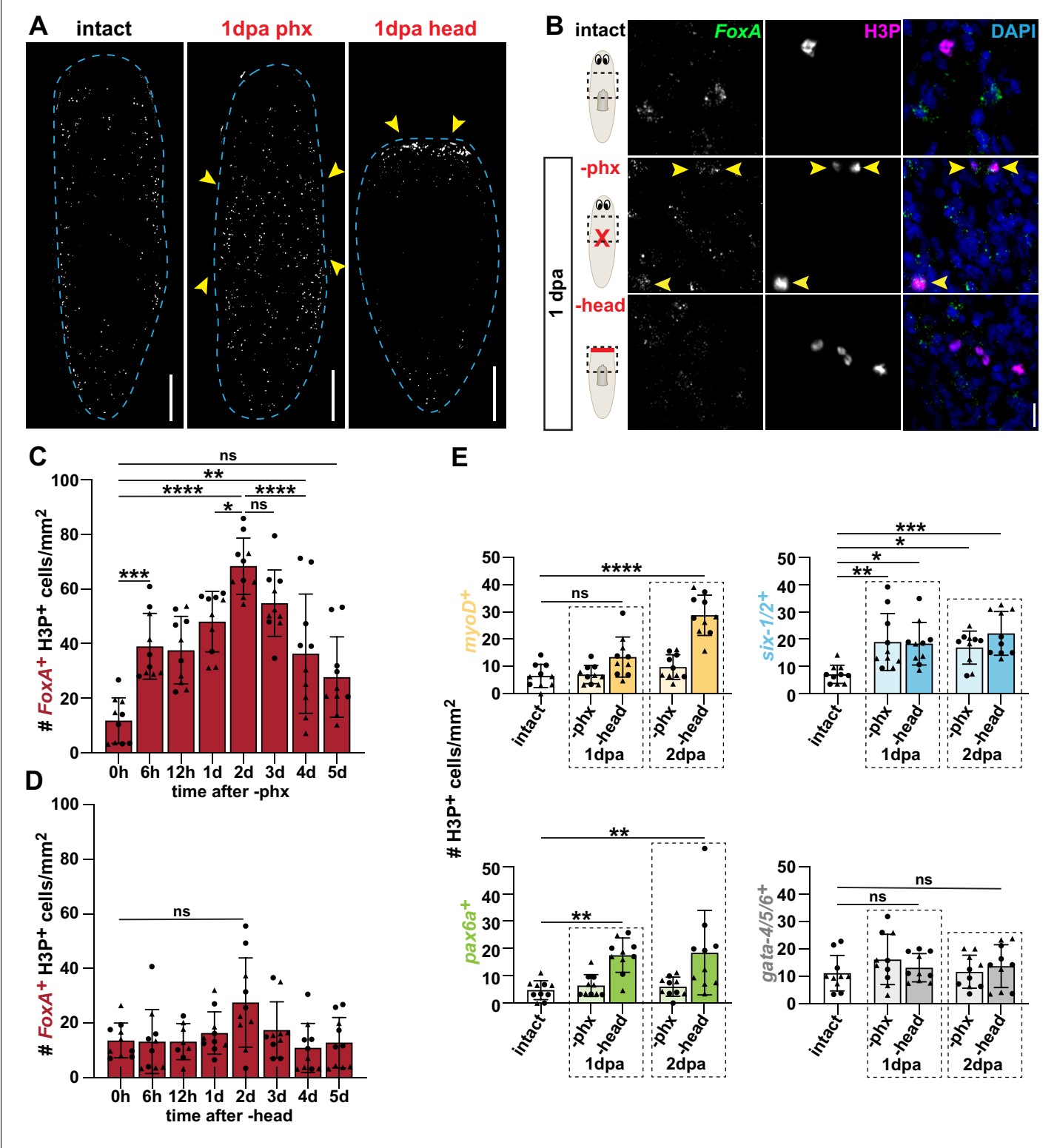

**Figure 3.** Pharynx loss selectively induces mitosis of pharynx progenitors. (**A**) Whole-mount images of animals stained with anti-phosphohistone H3 (H3P) in intact and injured animals, 1 day post-amputation (dpa). Dashed line outlines animal; arrows = areas of increased H3P; scale bars = 250 μm. (**B**) Confocal images of FISH for *FoxA* (green) and H3P antibody (magenta) in intact and injured animals, 1 day post-amputation. Images are partial projections of a portion of the area outlined by dashed boxes. DAPI = DNA (blue); arrows = double-positive cells; scale bar = 10 μm. (**C**) Number of *FoxA*+ H3P+ cells at indicated times after pharynx amputation in the area outlined by dashed boxes in B. (**D**) Number of *FoxA*+ H3P+ cells at indicated

*Figure 3 continued on next page*

Figure 3 continued

times after head amputation in the area outlined by dashed boxes in B. (E) Number of cells double-positive for H3P and the indicated progenitor marker in the area outlined by dashed boxes in B. For H3P quantification, the entire pre-pharyngeal region was analyzed over 30 z-sections, as represented by dashed boxes in B, and normalized to area. Graphs represent mean ± SD; symbols = individual animals; shapes distinguish biological replicates and; *, p≤0.05; **, p≤0.01; ***, p≤0.001; ****, p≤0.0001, one-way ANOVA with Tukey test. Raw data can be found in *Figure 3—source data 1*.

The online version of this article includes the following source data and figure supplement(s) for figure 3:

**Source data 1.** Quantification of H3P$^+$ cells in *Figure 3C–E*.

**Figure supplement 1.** Pharynx loss selectively increases the number of mitotic pharynx progenitors in proportion to all stem cells.

**Figure supplement 1—source data 1.** Quantification of H3P$^+$ cells in *Figure 3—figure supplement 1A and C*.

**Figure supplement 2.** Division of non-pharyngeal progenitors is not triggered by pharynx loss.

**Figure supplement 3.** Amputation-induced division of organ-specific progenitors is localized to wounds.

**Figure supplement 3—source data 1.** Quantification of H3P$^+$ cells in *Figure 3—figure supplement 3B–D*.

**Figure supplement 4.** Eye progenitors do not divide in response to amputation.

---

cells at various times after pharynx amputation in the pre-pharyngeal region (*Figure 3B*). Division of pharynx progenitors increased within 6 hr of pharynx amputation, peaked within 2 days, and returned to homeostatic levels by 5 days after amputation (*Figure 3C*). Despite an overall increase in H3P$^+$ stem cells 1 day after head amputation (*Figure 3A*; *Baguñà, 1976*), numbers of H3P$^+$ pharynx progenitors did not correspondingly increase (*Figure 3B,D*). Analysis of H3P$^+$ pharynx progenitors relative to all dividing stem cells in the same prepharyngeal region sustained this dichotomy (*Figure 3—figure supplement 1A*). Interestingly, we were able to detect what appeared to be instances of both symmetric and asymmetric distribution of *FoxA* in cells undergoing anaphase (*Figure 3—figure supplement 1B*). Together, these data show that pharynx progenitors are selectively stimulated to divide in response to pharynx loss.

In addition to the selective division of excretory system progenitors that occurs after RNAi depletion of excretory tissues (*Thi-Kim Vu et al., 2015*), progenitors in the epidermal lineage have also been shown to divide following head amputation (*van Wolfswinkel et al., 2014*). Therefore, we tested whether non-pharyngeal progenitors are selectively stimulated to divide 1 and 2 days after head amputation. Although the kinetics of each differed slightly, excretory system (*six1/2$^+$*), nervous system (*pax6a$^+$*), and muscle (*myoD$^+$*) progenitor division increased 2 days after head amputation, while intestinal (*gata-4/5/6$^+$*) progenitors did not. Additionally, division of nervous system and muscle progenitors increased only after head but not pharynx amputation (*Figure 3E*, *Figure 3—figure supplement 2A–D*). Analysis of these dividing progenitors relative to all H3P$^+$ stem cells recapitulated these results, with the exception of *six-1/2$^+$* excretory progenitors, which did not proportionally increase after either amputation (*Figure 3—figure supplement 1C*). Despite minor differences of each progenitor, the overall trend supports the notion that loss of non-pharyngeal tissues triggers division of stem cells expressing non-pharyngeal progenitor markers while in most cases, pharynx loss does not. This mitotic response appears again to be local, as we did not observe increased division of any of these organ-specific progenitors in tail regions, distant from wounds (*Figure 3—figure supplement 3A–D*). Even after head amputation, we were unable to detect any dividing eye progenitors (*ovo$^+$H3P$^+$*) (*Figure 3—figure supplement 4A*), unless we stalled mitotic exit with the microtubule destabilizing drug nocodazole. Following nocodazole treatment, we detected rare instances of H3P$^+$ eye progenitors; however, they did not increase after head amputation (*Figure 3—figure supplement 4B*). Therefore, while eye progenitors do divide, their division dynamics do not seem to be affected by injury, similar to intestinal progenitors (*Figure 3E*, *Figure 3—figure supplement 2D*). This finding suggests that there may be other eye and intestinal progenitors upstream of those expressing *ovo* or *gata-4/5/6*, or that expansion of these progenitors may occur via transcriptional upregulation.

## Stem cell division within a critical window is required for pharynx regeneration

In planarians, pulse-chase experiments using thymidine analogs have shown that cell division contributes to the production of regenerated tissues (*Cowles et al., 2013*; *Eisenhoffer et al., 2008*; *Forsthoefel et al., 2011*; *LoCascio et al., 2017*; *Newmark and Sánchez Alvarado, 2000*;

*Wagner et al., 2011*). Our results above identified an elevation in pharynx progenitor division within 2 days after pharynx removal (*Figure 3C*) that correlates with cell cycle entry of stem cells destined for missing pharynx tissue 1 day after amputation (*Figure 1E,F*). Together, these results define a window of 1–2 days after amputation in which pharynx progenitor division may selectively contribute to pharynx regeneration. Because this timeframe directly precedes the expansion of pharynx progenitors 3 days after pharynx amputation (*Figure 2C*), we hypothesized that stem cell division increases to specifically generate the progenitors that are necessary for pharynx regeneration.

To test this possibility, we blocked stem cell division with nocodazole, which induces a metaphase arrest with as little as 24 hr of exposure, resulting in the accumulation of mitotic (H3P$^+$) stem cells (*Figure 4A*; *Grohme et al., 2018*; *Molinaro et al., 2021*; *van Wolfswinkel et al., 2014*). To specifically inhibit mitosis 1–2 days after pharynx amputation, we soaked animals in nocodazole for 24 hr, beginning 1 day after pharynx amputation (*Figure 4B*). We then assayed pharynx regeneration via recovery of feeding behavior. Animals treated with nocodazole for 24 hr had drastic delays in recovery of feeding, compared to DMSO-treated controls, with only 50% of worms regaining the ability to eat within 20 days, and 100% within 32 days of amputation (*Figure 4C*). To verify that nocodazole treatment under these conditions delayed pharynx regeneration, we examined pharynx anatomy with the marker *laminin*, which is strongly expressed in the mouth and pharynx, and weakly expressed in the body where the pharynx attaches (*Adler et al., 2014*; *Cebrià and Newmark, 2007*). As expected, while residual *laminin* expression was retained within the body, animals treated with nocodazole 1–2 days after pharynx amputation completely lacked a pharynx with its characteristic layered structure 7 days after amputation and sustained severe defects even up to 14 days (*Figure 4D*). Treatment with nocodazole for a full 48 hr, beginning immediately after amputation (*Figure 4B*), did not exacerbate the delay in feeding ability or defects in pharynx anatomy (*Figure 4C,D*). These findings suggest that stem cell division outside the 1–2 day window has a minor contribution to pharynx regeneration. To verify this, we soaked animals in nocodazole for 24 hr increments surrounding this 1–2 day window, beginning either immediately, or 2 days after pharynx amputation, which recovered feeding behavior at a similar rate as controls (*Figure 4—figure supplement 1A,B*). Further, animals treated from 0 to 1 days after amputation had only minor defects in pharynx anatomy, while those treated 2–3 days after amputation were normal (*Figure 4—figure supplement 1C*). Therefore, we conclude that stem cell division within a critical window of 1–2 days after amputation fuels the majority of pharynx regeneration.

To test whether stem cell division within this critical window generates pharynx progenitors, we exposed animals to nocodazole 1–2 days after amputation, and analyzed the impact on *FoxA*$^+$ stem cells. First, we verified that this treatment caused an extensive increase in H3P$^+$ pharynx progenitors 2 days after amputation (*Figure 4—figure supplement 2A,B*), illustrating that they were arrested in mitosis. Second, we performed double FISH for *FoxA* and *piwi-1* 3 days after pharynx removal and found that nocodazole treatment caused a dramatic decrease in pharynx progenitors compared to controls (*Figure 4E,F*). Importantly, intact animals treated similarly with nocodazole showed no difference in the abundance of pharynx progenitors compared to controls (*Figure 4—figure supplement 2C,D*). Therefore, perturbing stem cell division during this brief window specifically impacts the production of pharynx progenitors during regeneration. To determine if stem cell division during other times contributed to pharynx progenitor production, we again exposed animals to nocodazole for 0–1 and 2–3 days after pharynx amputation. While we observed some defects in the production of pharynx progenitors, they were more subtle than those in animals treated for 1–2 days after amputation (*Figure 4—figure supplement 2E*). Together, our data show that stem cell division in a critical window of 1–2 days after amputation produces pharynx progenitors that are likely essential for pharynx regeneration.

## ERK phosphorylation is required to produce pharyngeal progenitors

The mitogen activated protein (MAP) kinase pathway drives proliferation and differentiation during development and regeneration in many organisms (*Ghilardi et al., 2020*; *Patel and Shvartsman, 2018*). In planarians, phosphorylation of the MAP kinase ERK is the earliest known injury-induced signal required for regeneration. ERK activity regulates broad stem cell proliferation and is required for transcriptional changes that drive axial repatterning (*Owlarn et al., 2017*; *Tasaki et al., 2011*). To determine whether pharynx loss also induces ERK phosphorylation, we performed a western blot with an antibody against phosphorylated ERK (pERK). pERK increased 15 min after pharynx

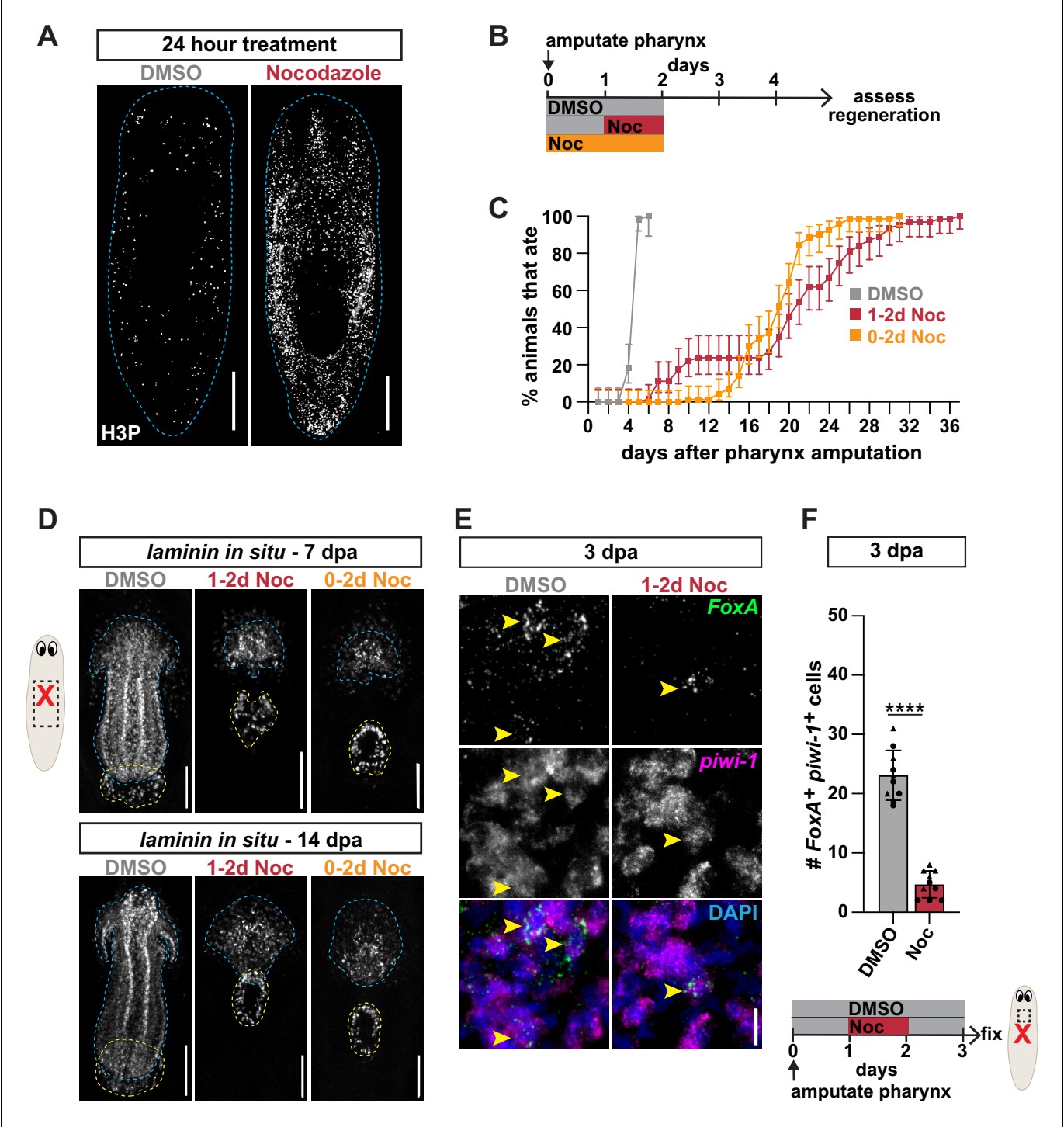

**Figure 4.** Stem cell division within a critical window is required for pharynx regeneration. (**A**) Whole-mount images of intact animals stained with H3P after treatment with DMSO or nocodazole for 24 hr. Dashed line outlines animal; scale bars = 250 μm. (**B**) Schematic of nocodazole treatment relative to pharynx amputation for graph in C and images in D. (**C**) Proportion of animals capable of feeding after pharynx amputation, treated as indicated in B and assayed daily. Error bars represent ± 95% confidence intervals. n ≥ 54 animals from three independent experiments. (**D**) Whole-mount FISH for the pharynx marker *laminin* 7 and 14 days post-pharynx amputation (dpa) in animals treated as indicated in B. Dashed blue line outlines pharynx; dashed yellow line outlines mouth; scale bars = 100 μm. n ≥ 23 animals from three independent experiments. (**E**) Confocal images of FISH for *FoxA* (green) and *piwi-1* (magenta) 3 days post-pharynx amputation in animals treated with DMSO or nocodazole, 1 day after amputation for 24 hr. DAPI = DNA (blue); arrows = double-positive cells; scale bar = 10 μm. (**F**) Number of *FoxA*⁺ *piwi-1*⁺ cells quantified in animals after indicated treatments (schematic) in the

*Figure 4 continued on next page*

Figure 4 continued

area outlined by dashed box in cartoon. Graph represents mean ± SD; symbols = individual animals; shapes distinguish biological replicates and; ****, p≤0.0001, unpaired t-test. Raw data can be found in *Figure 4—source data 1*.

The online version of this article includes the following source data and figure supplement(s) for figure 4:

**Source data 1.** Raw data for feeding assay (*Figure 4C*) and quantification of *piwi-1*+ cells (*Figure 4F*).

**Figure supplement 1.** Stem cell division outside a critical window is not required for pharynx regeneration.

**Figure supplement 1—source data 1.** Raw data for feeding assay in *Figure 4—figure supplement 1B*.

**Figure supplement 2.** Inhibiting stem cell division for 24 hr reduces pharynx progenitors during regeneration.

**Figure supplement 2—source data 1.** Quantification of H3P+ cells (*Figure 4—figure supplement 2B*), and *piwi-1*+ cells (*Figure 4—figure supplement 2D and E*).

amputation and returned to baseline levels within 6 hr (*Figure 5A*). Therefore, similar to other injuries (*Owlarn et al., 2017*), pharynx amputation also activates ERK by phosphorylation soon after injury.

Exposing animals to PD0325901 (PD), an inhibitor of the upstream ERK-activating MEK kinase, blocks ERK phosphorylation and permanently inhibits regeneration following substantial anterior tissue removal (*Owlarn et al., 2017*). To determine whether ERK is also required for pharynx regeneration, we exposed animals to PD for 5 days immediately after pharynx amputation and then assayed feeding behavior (*Figure 5B,C*). While DMSO-treated control animals regained the ability to feed within 7 days, animals treated with PD from 0 to 5 days after amputation had substantial delays in feeding, with 50% of worms feeding by day 13 and all worms feeding by day 29 (*Figure 5C*). Depending on the timing, delaying administration of MEK inhibitors relative to amputation partially or completely rescues anterior regeneration, and suggests that ERK acts within the first day of regeneration (*Owlarn et al., 2017*). To pinpoint when ERK signaling is important for pharynx regeneration, we delayed PD exposure for 1 or 2 days after pharynx amputation, and again assayed feeding (*Figure 5B,C*). Animals exposed 1 day after pharynx amputation (1–6 days) had delayed feeding ability, similar to those treated immediately after amputation (0–5 days), suggesting that ERK activity within the first day of amputation is dispensable for pharynx regeneration. Animals exposed 2 days after amputation (2–7 days) regained the ability to feed at rates similar to controls (*Figure 5C*), indicating that ERK is essential for regeneration within the first 2 days after pharynx amputation. Therefore, ERK likely acts primarily between 1 and 2 days after pharynx amputation. This timing occurs after the increase in pERK following injury has already subsided (*Figure 5A*), suggesting that pharynx regeneration may be facilitated by homeostatic levels of ERK signaling instead of its injury-induced high level activation.

To verify that the inability of ERK-inhibited animals to feed was caused by defects in regeneration, we analyzed pharynx anatomy 7 days after pharynx amputation with FISH for *laminin*. Animals exposed to PD for 5 days, beginning 0 or 1 day after amputation, lacked a pharynx, while pharynges in animals exposed beginning 2 days after amputation were comparable to controls (*Figure 5D*), mirroring the results of our feeding assay (*Figure 5C*). PD-exposed animals eventually regenerated a pharynx within about 2 weeks (*Figure 5C*, *Figure 5—figure supplement 1A*), suggesting that ERK inhibition does not permanently block pharynx regeneration. Therefore, we confirmed the effects of ERK inhibition by repeating pharynx regeneration experiments with UO126 (UO), another potent, but structurally independent, MEK inhibitor and observed the same outcomes (*Figure 5—figure supplement 1B–D*). Furthermore, exposure to PD or UO eliminated pERK after pharynx amputation on a western blot (*Figure 5—figure supplement 1E*) and also permanently blocked regeneration after extensive anterior tissue removal, for up to 70 days (*Figure 5—figure supplement 1F*). Together, these results define a window 1–2 days after amputation in which ERK activity is required for pharynx regeneration.

Soon after amputation, ERK signaling is required for upregulation of several genes including *follistatin (fst)* (*Owlarn et al., 2017*), which accelerates regeneration by inhibiting *activin-1* and *-2* (*Gaviño et al., 2013*; *Roberts-Galbraith and Newmark, 2013*; *Tewari et al., 2018*). Like ERK inhibition, *fst(RNAi)* prevents regeneration following substantial anterior tissue removal (*Owlarn et al., 2017*; *Tewari et al., 2018*). However, if less tissue is removed, the requirement for *fst* diminishes (*Tewari et al., 2018*). Likewise, when we amputated animals pre-pharyngeally and maintained them to PD for 5 days, head regeneration was initially delayed in 100% of animals 7 days after amputation,

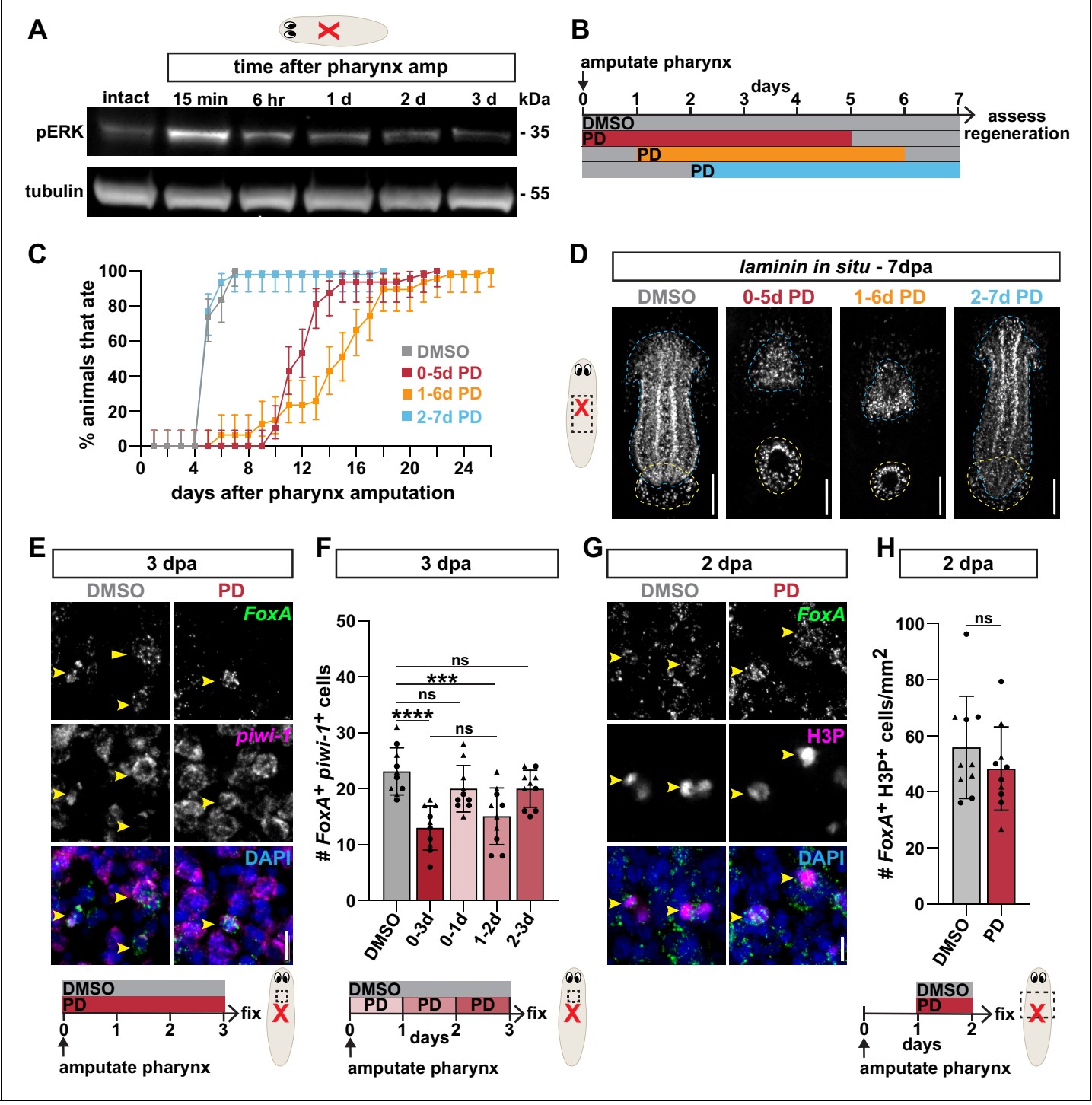

**Figure 5.** ERK phosphorylation is required to produce pharyngeal progenitors. (**A**) Western blot for phosphorylated ERK (pERK) and tubulin (loading control) in intact animals and at the indicated times after pharynx amputation. (**B**) Schematic of PD0325901 (PD) exposure relative to pharynx amputation for graph in C and images in D. (**C**) Proportion of animals capable of feeding after pharynx amputation, treated as indicated in B and assayed daily. Error bars represent ± 95% confidence intervals. n ≥ 47 animals from three independent experiments. (**D**) Whole-mount FISH for the pharynx marker *laminin* 7 days post-pharynx amputation (dpa) in animals treated as indicated in B. Dashed blue line outlines pharynx; dashed yellow line outlines mouth. Scale bars = 100 µm. n ≥ 18 animals from two independent experiments. (**E**) Confocal images of FISH for *FoxA* (green) and *piwi-1* (magenta) 3 days post-pharynx amputation in animals treated with DMSO or PD (schematic). DAPI = DNA (blue); dashed box = region imaged; arrows = double-positive cells; scale bar = 10 µm. (**F**) Number of *FoxA*⁺ *piwi-1*⁺ cells 3 days post-pharynx amputation after indicated treatments (schematics E and F). (**G**) Confocal images of *FoxA* FISH (green) and H3P antibody (magenta) 2 days post-pharynx amputation in animals treated with

*Figure 5 continued on next page*

*Figure 5 continued*

DMSO or PD, 1 day after amputation for 24 hr. DAPI = DNA (blue); arrows = double-positive cells; scale bar = 10 μm. (H) Number of *FoxA*$^+$ H3P$^+$ cells 2 days post-pharynx amputation in animals treated with DMSO or PD (schematic). Bar graphs represent mean ± SD; symbols = individual animals; shapes distinguish biological replicates and; ***, p≤0.001; ****, p≤0.0001; one-way ANOVA with Tukey test (F), unpaired t-test (H). Raw data can be found in *Figure 5—source data 1*.

The online version of this article includes the following source data and figure supplement(s) for figure 5:

**Source data 1.** Original, uncropped images of western blots in *Figure 5A*.
**Source data 2.** Raw data for feeding assay (*Figure 5C*), quantification of *piwi-1*$^+$ cells (*Figure 5F*) and H3P$^+$ cells (*Figure 5H*).
**Figure supplement 1.** MEK inhibitors U0126 and PD0325901 prevent ERK phosphorylation and tissue regeneration.
**Figure supplement 1—source data 1.** Raw data for feeding assay in *Figure 5—figure supplement 1C*.
**Figure supplement 1—source data 2.** Original, uncropped images of western blots in *Figure 5—figure supplement 1E*.
**Figure supplement 2.** ERK-dependent pharynx regeneration is independent of *follistatin*.
**Figure supplement 2—source data 1.** Quantification of feeding behavior in *Figure 5—figure supplement 2B*.
**Figure supplement 3.** Inhibiting ERK phosphorylation reduces pharynx progenitors during regeneration.
**Figure supplement 3—source data 1.** Raw data for quantification of *piwi-1*$^+$ cells (*Figure 5—figure supplement 3A and B*), feeding assay (*Figure 5—figure supplement 3C*), and H3P$^+$ cells (*Figure 5—figure supplement 3D and E*).

but eventually occurred in 88% of animals within 2 weeks (*Figure 5—figure supplement 1G*), on a similar timeline as pharynges (*Figure 5C,D*, *Figure 5—figure supplement 1A*). Therefore, ERK may mediate pharynx regeneration via *fst* expression, resulting in a short-lived block of pharynx regeneration. To test this, we analyzed *fst* expression, which increases within 6 hr of head amputation (*Gaviño et al., 2013*), but not until 24 hr after pharynx amputation (*Figure 5—figure supplement 2A*), suggesting that injury-induced ERK activation (*Figure 5A*) may not always coincide with *fst* upregulation. Despite upregulation of *fst* expression after pharynx amputation, *fst(RNAi)* animals regained the ability to feed at a normal rate (*Figure 5—figure supplement 2B and C*), indicating that *fst* is not required to accelerate pharynx regeneration. Therefore, although pharynx loss eventually induces *fst* expression, regulation of pharynx regeneration via ERK is independent of *fst*.

Our data suggests that ERK activity is required 1–2 days after pharynx amputation, just prior to the emergence of pharynx progenitors 3 days after amputation. We hypothesized that ERK may promote the production of these progenitors, which we tested by maintaining animals in PD for 3 days following pharynx amputation (*Figure 5E*). PD exposure significantly inhibited the increase in *FoxA*$^+$-*piwi-1*$^+$ cells typically observed after pharynx amputation (*Figure 5E,F*). Importantly, intact animals treated with PD for 3 days showed no difference in the abundance of pharynx progenitors as compared to controls (*Figure 5—figure supplement 3A*), indicating that ERK activity for this duration is not necessary for maintaining pharynx progenitors during homeostasis. Therefore, the decrease in pharynx progenitors following pharynx amputation and PD treatment is likely due to their reduced production, rather than altered survival. To determine when ERK promotes pharynx progenitor production, we exposed animals to PD in 24 hr increments during this 3-day window (*Figure 5F*). While PD treatment starting 0 or 2 days after pharynx amputation had no effect on pharynx progenitors, treatment starting 1 day after amputation significantly inhibited pharynx progenitor increase, similar to those treated for three full days (*Figure 5F*). Similar experiments with UO yielded the same outcomes (*Figure 5—figure supplement 3A,B*), demonstrating that ERK activity 1–2 days after pharynx amputation promotes the production of pharynx progenitors during regeneration.

To test whether PD-mediated inhibition of pharynx progenitor production following pharynx loss has long-term consequences on pharynx regeneration, we exposed animals to PD for 0–3 days and 1–2 days after amputation, and assayed feeding behavior. These animals had delays in feeding, although less severe than those treated for five full days (*Figure 5—figure supplement 3C*), presumably because animals start recovering after drug washout. Meanwhile, those treated for 0–1 day after pharynx amputation recovered feeding ability at normal rates (*Figure 5—figure supplement 3C*), illustrating that PD exposure that does not affect the production of pharynx progenitors does not delay regeneration. Together, these results indicate that ERK activity 1–2 days after amputation promotes pharynx regeneration by contributing to the increase in pharynx progenitors 3 days after amputation.

Because ERK signaling regulates broad stem cell division associated with missing tissue in planaria (*Owlarn et al., 2017*), the decrease in pharynx progenitor production after drug exposure could be

a result of reduced stem cell division. To determine if amputation-induced pharynx progenitor division depends on ERK activity, we exposed animals to MEK inhibitors 1–2 days after pharynx amputation, and analyzed $FoxA^+$ H3P$^+$ stem cells. Neither PD or UO exposure substantially impacted the number of dividing pharynx progenitors as compared to controls (*Figure 5G,H*, *Figure 5—figure supplement 3D*), despite an overall decrease in H3P$^+$ stem cells in the same animals (*Figure 5—figure supplement 3E*). These data indicate that ERK signaling is not required for the selective increase in pharynx progenitor division induced by pharynx loss. Because the timing of ERK's requirement for increasing pharynx progenitors overlaps with when stem cell division is required (*Figure 4C–F*) 1–2 days after pharynx removal, ERK signaling is unlikely to regulate initiation of pharynx progenitor division. Instead, ERK likely promotes pharynx progenitor production and regeneration by regulating *FoxA* expression and stem cell differentiation.

## Stem cell division and ERK phosphorylation are not required for eye regeneration following selective removal

Unlike pharynx removal, selective removal of the eye does not increase stem cell division or expansion of eye-specific progenitors. Instead, following resection, the eye regenerates by passive homeostatic turnover that is not regulated by the presence or absence of the eye (*LoCascio et al., 2017*). Therefore, we speculated that eye regeneration, following specific removal, may not have the same requirements as the pharynx. To test whether eye regeneration relies on stem cell division and ERK signaling, we performed eye-specific resections and immediately exposed animals to either nocodazole or MEK inhibitors (*Figure 6A*). We monitored eye regeneration with FISH for *ovo* and the eye-specific marker *opsin Sánchez Alvarado and Newmark, 1999*. We confirmed that eye tissue was successfully removed by the absence of *ovo$^+$ opsin $^+$* photoreceptors immediately after surgery (*Figure 6B*). Exposure to either nocodazole for 2 days, or MEK inhibitors for 5 days, did not impact eye regeneration (*Figure 6B,C*), despite a complete block of pharynx regeneration under these conditions (*Figure 4D*, *Figure 5D*, *Figure 5—figure supplement 1D*). Our results show that, unlike pharynx regeneration, eye regeneration does not require stem cell division or ERK activity after selective removal. Instead, much like the maintenance of pharynx progenitors in intact animals (*Figure 4—figure supplement 2D*, *Figure 5—figure supplement 3A*), eye regeneration is unaffected by drug treatments. Therefore, it is possible that pre-existing eye progenitors, or more likely, the continued production of homeostatic levels of eye progenitors (*LoCascio et al., 2017*) is sufficient for eye regeneration following selective removal.

Previous studies have shown that eye regeneration may occur in alternative ways depending on the context of the wound. While eye-specific resections do not invoke typical injury responses, more severe injuries stimulate the expansion of *ovo$^+$* eye progenitors (*Lapan and Reddien, 2012*; *LoCascio et al., 2017*). Because wounds generated from eye resection do not stimulate the same response as more severe injuries, their repair may not depend on the same mechanisms facilitated by proliferation and ERK signaling. Therefore, we tested whether stem cell division and ERK activity are required for eye regeneration after head amputation (*Figure 6A*). In controls, *ovo$^+$ opsin $^+$*-photoreceptors re-emerged 7 days after amputation (*Figure 6D,E*; *Lapan and Reddien, 2011*). By contrast, animals treated with either nocodazole for 2 days or ERK inhibitors for 5 days failed to regenerate eyes, despite the presence of *ovo$^+$* eye progenitors (*Figure 6D,E*). We conclude that unlike pharynx regeneration, eye regeneration only requires stem cell division and ERK signaling in the context of more severe injuries.

## Discussion

In this study, to evaluate the contribution of stem cells to tissue regeneration, we challenged stem cells by inflicting different types of injuries to planarians. Analysis of organ-specific progenitors after various wounds highlighted shifts in the abundance of these stem cell populations based on the presence or absence of specific organs. We uncovered a mechanism that targets regeneration towards missing tissues, fueled by the division and subsequent expansion of organ-specific progenitors required for regeneration. In particular, we found that pharynx loss induces a selective increase in pharynx progenitor division, 1–2 days after amputation, followed by an ERK-dependent expansion of pharynx progenitors that are channeled towards pharynx regeneration (*Figure 7*). These findings

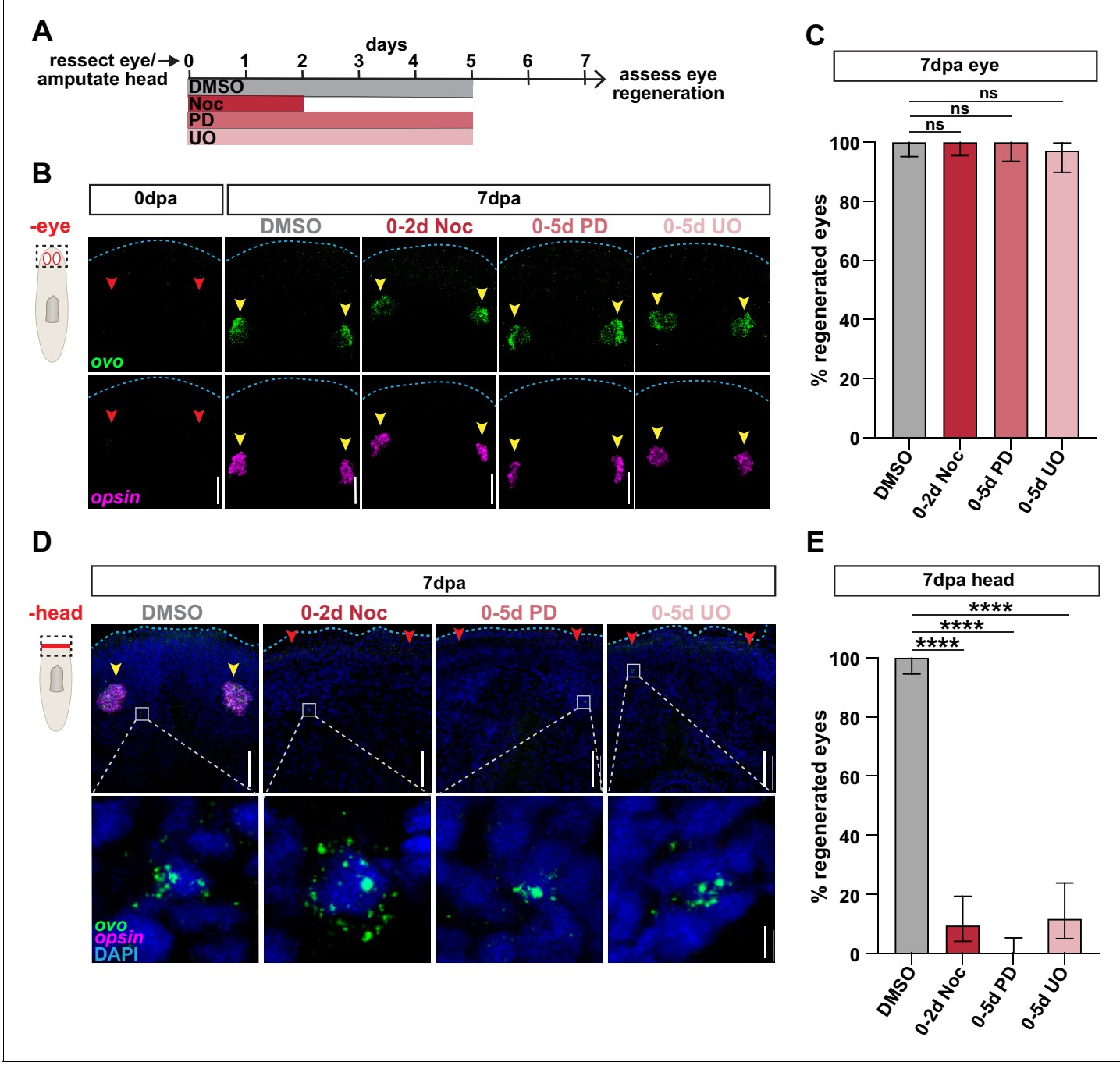

**Figure 6.** Stem cell division and ERK activation are not required for eye regeneration following selective removal. (**A**) Schematic of drug treatment relative to injuries for B-E. (**B**) Confocal images of FISH for *ovo* (green) and *opsin* (magenta) immediately (0dpa), or 7 days post-eye resection in animals treated as in A. Dashed box in cartoon represents area imaged. Blue dashed line outlines anterior edge of worm; red arrows = missing eyes; yellow arrows = regenerated eyes; scale bars = 50 μm. (**C**) Proportion of *ovo*⁺ *opsin*⁺ eyes that regenerated 7 days post-eye resection. n ≥ 34 animals from three independent experiments. (**D**) Confocal images of FISH for *ovo* (green) and *opsin* (magenta) 7 days post-head amputation in animals treated as in A. Dashed box in cartoon represents area in top images. DAPI = DNA (blue); blue dashed line outlines anterior edge of worm; red arrows = missing eyes; yellow arrows = regenerated eyes; scale bars = 50 μm. Bottom = zoom of *ovo*⁺ cells from the regions outlined by gray boxes in top images. Scale bars = 5 μm. (**E**) Proportion of *ovo*⁺ *opsin*⁺ eyes that regenerated 7 days post-head amputation. n ≥ 32 animals from three independent experiments. In graphs, error bars = ± 95% confidence intervals and; ****, p≤0.0001; Fisher's Exact Test. Raw data can be found in *Figure 6—source data 1*.

The online version of this article includes the following source data for figure 6:

**Source data 1.** Quantification of eye regeneration in *Figure 6C and E*.

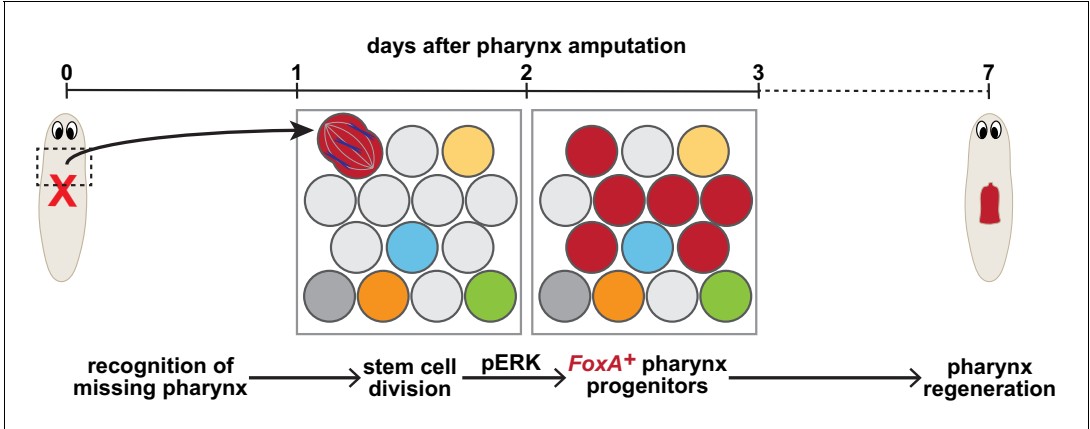

**Figure 7.** Model for targeted pharynx regeneration. Soon after pharynx loss, stem cells recognize the pharynx is missing and target regeneration toward the pharynx by selectively inducing division of existing *FoxA*-expressing stem cells (red), or expression of *FoxA* in division-competent stem cells, within 1–2 days. This division drives an ERK-dependent increase in pharynx progenitors 3 days after pharynx loss, which is required for pharynx regeneration.

suggest that in many cases, stem cells can sense the identity of missing tissues to launch their targeted regeneration.

## Modes of regeneration: targeted or non-targeted?

Previous work has suggested two potential models underlying planarian regeneration. One group proposed a non-targeted model, in which stem cells broadly incorporate into both damaged and undamaged tissue, dependent on indiscriminate amplification of progenitors triggered by nearby wounds (*LoCascio et al., 2017*). On the other hand, a targeted model suggests that stem cells amplify organ-specific progenitors in response to perturbations to organ tissues (*Thi-Kim Vu et al., 2015*). By labeling proliferating stem cells with F-ara-EdU at different times after amputation, we have uncoupled the contribution of these two mechanisms to the production of regenerated tissues. Depending on the type of injury, cells generated soon after amputation are channeled into all nearby tissues, even undamaged ones. However, those generated 1 day after amputation are selectively targeted for missing tissues. Importantly, the timing of this targeted mechanism overlaps with an essential peak in pharynx progenitor division, revealing that this mechanism generates organ progenitors necessary for the replacement of missing tissues.

The cells that regenerate the pharynx are primarily generated through this targeted mechanism, mediated by stem cell division and ERK signaling between 1and 2 days after amputation (*Figure 7*). Smaller lesions, such as eye resections, do not stimulate a proliferative wound response (*LoCascio et al., 2017*), nor do they depend on stem cell division and ERK signaling for subsequent regeneration. A previous study proposed that regeneration initiation requires signals regulated by both injury and tissue loss (*Owlarn et al., 2017*). If eye removal is not detected as an injury, and is insufficient to trigger a 'missing tissue signal', this could explain why eye regeneration, following resection, happens passively via homeostatic cellular turnover in the regenerating eye (*LoCascio et al., 2017*). Further, while surgical removal of the pharynx does increase production of neural tissue (*LoCascio et al., 2017*), selective chemical pharynx removal does not, suggesting that non-targeted regeneration may require injury to specific types of tissues, such as body wall muscle or epithelia, in addition to tissue loss. Therefore, multiple avenues lead to regeneration: passive homeostatic turnover in the absence of typical injury responses, and both targeted and non-targeted mechanisms that increase cellular production when tissue is lost.

Our work highlights several key features that are important to consider for future analysis of regeneration. In particular, depending on the size and severity of the wound, initial injury signals may vary, triggering a differential requirement for stem cell division and regulatory signaling pathways. Our results underscore the distinct requirements for proliferation and ERK activity in regeneration of eyes and the pharynx. Also, the timing of experimentation and analysis should be deliberate, as we show that F-ara-EdU-labeled stem cells are differentially incorporated into regenerating and

non-regenerating tissues depending on when they are labeled relative to injury. Another key consideration is the intrinsic heterogeneity of cell populations that are analyzed. Stem cells identified by *piwi-1* and H3P staining encompass cells with different potencies and differentiation states (*van Wolfswinkel et al., 2014*; *Zeng et al., 2018*), which may respond uniquely to the changing environment of a regenerating animal. Therefore, broad analysis of stem cells lacks the resolution required to tease apart the intricacies involved in coordinating regeneration. Restricting our analysis to organ progenitors and even further to those that are actively dividing narrows this focus. However, our observation of the asymmetric and symmetric segregation of *FoxA* in dividing cells suggests that heterogeneity exists even within these subsets of stem cells. Resolving the specific stem cells responsible for driving different modes of regeneration, and when cell fate is established in them, will be an exciting area for future work.

## ERK signaling plays multiple roles during regeneration

Phosphorylation of ERK promotes regeneration in many animals (*DuBuc et al., 2014*; *Wan et al., 2012*; *Yun et al., 2014*). In planaria, ERK has been implicated as a regeneration trigger, as it is briefly activated by phosphorylation within minutes of injury and is required for wound-induced transcription, stem cell differentiation and broad stem cell proliferation (*Owlarn et al., 2017*; *Tasaki et al., 2011*). ERK also functions to re-establish axial patterning during regeneration (*Owlarn et al., 2017*; *Umesono et al., 2013*), which depends on a network of positional cues that are expressed in muscle cells throughout the body (*Lander and Petersen, 2016*; *Scimone et al., 2016*; *Witchley et al., 2013*). Because tissue removal from the body requires re-establishment of these positional cues for regeneration to proceed (*Rink, 2018*), it has been difficult to distinguish ERK's roles in organ regeneration.

Unlike amputations to the body, pharynx removal does not broadly disrupt positional cues, which has allowed us to pinpoint a distinct role for ERK in organ regeneration. Both ERK activity and stem cell division act simultaneously, but independently, 1–2 days after pharynx loss, to drive the expansion of pharynx progenitors 3 days after amputation. These events occur after the injury-induced increase in pERK has subsided. Also, ERK activity is dispensable for pharynx progenitor division. Taken together, these results suggest that an ERK-independent signal triggers division of $FoxA^+$ stem cells, and that ERK acts later during organ regeneration to facilitate stem cell differentiation or maintain cell fate.

Among the ERK-dependent wound-induced genes is *follistatin (fst)* (*Owlarn et al., 2017*), which promotes regeneration in ways similar to ERK (*Tewari et al., 2018*). Interestingly, both ERK and *fst* are absolutely essential for regeneration following substantial anterior tissue removal, but become less so if smaller amounts of tissue are removed. This variability may be due to the extent of axial patterning disruption induced by different amputations. In fact, the inability of *fst(RNAi)* animals to regenerate is entirely dependent on their failure to reset positional information, as inhibition of *wnt* signals that restrict head formation rescue these regeneration defects (*Tewari et al., 2018*). Therefore, it is likely that injury-induced *fst* expression and ERK phosphorylation primarily regulate regeneration initiation by establishing repatterning rather than triggering stem cell behaviors that directly contribute to organ regeneration. ERK inhibition reduces broad stem cell division after tissue removal (*Owlarn et al., 2017*), but not the specific increase in division of pharynx progenitors that accompanies pharynx loss, suggesting that not all injury-induced stem cell behaviors may be critical for regeneration. Further, rescuing axial repatterning, and thus regeneration after *fst* knockdown, does not rescue defects in missing tissue-induced stem cell proliferation and apoptosis (*Tewari et al., 2018*). Therefore, the 'missing-tissue response' may encompass multiple events including patterning, broad stem cell division, and the generation of organ-specific progenitors that contribute independently to different aspects of regeneration.

Receptor tyrosine kinases such as the epidermal growth factor receptor (EGFR) and the fibroblast growth factor receptor (FGFR) have been shown to play critical roles in signaling upstream of ERK in many organisms (*Patel and Shvartsman, 2018*), making them intriguing candidates for potential regulators of regeneration. In planarians, *egfr-3* is required to activate ERK during regeneration (*Fraguas et al., 2017*) and is also involved in stem cell differentiation (*Fraguas et al., 2011*; *Lei et al., 2016*). Other studies have highlighted roles for the ligand *egf-4* and the receptors *egfr-1* and *egfr-5* in the differentiation of stem cells into brain, intestinal and excretory tissues, respectively (*Barberán et al., 2016*; *Fraguas et al., 2014*; *Rink et al., 2011*). Further, some FGFRL-Wnt circuits

restrict pharynx formation to the trunk region, possibly through regulation of *FoxA* expression (*Lander and Petersen, 2016*; *Scimone et al., 2016*). Whether any of the planarian EGF or FGF ligands or receptors similarly regulate the production of pharyngeal progenitors remains to be determined (*Cebrià et al., 2002*; *Ogawa et al., 2002*).

## Shifts in stem cell heterogeneity contribute to regeneration

By studying the dynamics of *FoxA* expression in stem cells after pharynx or head removal, we have uncovered shifts in stem cell heterogeneity that depend on the presence or absence of a particular organ. Intriguingly, we find that both the overall number of $FoxA^+$ pharynx progenitors, as well as those that are actively dividing, increase only after pharynx removal. Therefore, stem cells sense the absence of the pharynx and channel their proliferative output toward the population of stem cells required to regenerate it. Combined with our analysis of non-pharyngeal progenitor dynamics after different amputations, our results suggest that the heterogeneity of the stem cell population can be differentially deployed depending on the severity of the injury and the particular tissues that need repair.

Interestingly, removal of non-pharyngeal tissues like the head, does not increase pharynx progenitor proliferation but nevertheless results in increased F-ara $EdU^+$ cells within the pharynx, raising a conundrum regarding the source of these cells. One possibility is that other, non-$FoxA^+$ progenitors which have yet to be identified, may contribute to pharynx regeneration. However, no other tissue-specific transcription factors or proposed pharynx progenitor markers (*meis*, *twist*, or *dd_554*) appear to be required for pharynx regeneration (*Cowles et al., 2013*; *Scimone et al., 2014a*; *Zhu et al., 2015*). Alternatively, a recent study has suggested that planarian stem cells, even those expressing organ-specific transcription factors, may harbor a large degree of plasticity that allows fate switching between stem cell and progenitor types (*Raz et al., 2021*). While this hypothesis has not been tested in the context of injury, it is possible that stem cells generated soon after tissue loss could adopt a pharynx progenitor fate at various times over the course of regeneration, which would not necessarily generate a detectable increase in pharynx progenitors at any one time. It will be interesting to explore the potential of these cells in more detail when true lineage-tracing becomes possible in planarians.

Surprisingly, stem cell division in a narrow window, 1–2 days following pharynx amputation, is absolutely essential for pharynx regeneration, and coincides with the elevation of pharynx progenitor division that directly precedes their increase 3 days after pharynx amputation. The requirement for division in this brief moment after amputation suggests that stem cells detect tissue loss through transient signals regulated by injury. Indeed, a recent study has identified a population of potentially slow-cycling stem cells, reminiscent of reserve stem cells in mammals, that may be specifically induced to enter the cell cycle by tissue loss (*Bankaitis et al., 2018*; *Molinaro et al., 2021*). Whether this distinct population of stem cells contributes to the regeneration of particular organs is not known. Regulatory signals could either be produced upon injury to promote regeneration, or released from inhibitory cues that might emanate from organs when they are present (*Rink, 2018*; *Ziller-Sengel, 1967*; *Ziller-Sengel, 1965*). Intriguingly, a recent study characterizing transient amputation-induced transcriptional changes revealed that the majority of these changes occur within differentiated cell types (*Benham-Pyle et al., 2020*). The possibility of transient signals customized to particular organs and the ability of stem cells to readily sense them may explain how planarians exhibit such rapid and robust regeneration of all organs.

Cell fate acquisition can occur throughout the cell cycle (*Fichelson et al., 2005*; *Pauklin and Vallier, 2014*; *Soufi and Dalton, 2016*). The increase in *FoxA* expression in both actively dividing stem cells, and those outside of M-phase, does not pinpoint a particular time in the cell cycle where fate acquisition during regeneration might occur. Tissue loss could generate fleeting signals sensed by stem cells that influence them to adopt a specific cell fate during division to compensate for missing tissue. Alternatively, stem cells already expressing organ-specific markers may be poised to divide upon receiving such a signal, allowing them to quickly initiate regeneration upon exit of the cell cycle. Indeed, studies in human hepatoma cell lines have shown that FoxA1 remains attached to chromatin during mitosis, contributing to rapid activation of downstream targets following mitosis during liver differentiation (*Caravaca et al., 2013*). It will be interesting to explore these possibilities in future studies.

Mammalian homologs of *FoxA* were the first identified 'pioneer' transcription factors, characterized by their ability to engage closed chromatin and drive organogenesis (*Hsu et al., 2015*; *Iwafuchi-Doi and Zaret, 2016*; *Lam et al., 2013*; *Zaret and Mango, 2016*). This raises the possibility that pioneer factors may be viable *in vivo* targets for achieving regeneration of entire organs. In fact, overexpression of a related mammalian transcription factor, *FoxN*, is sufficient to drive regeneration of the thymus in mice (*Bredenkamp et al., 2014*). The increased proliferation of stem cells expressing *FoxA* after pharynx removal suggests that activation of pioneer factors may also drive organ regeneration in planaria. Other pioneer factors, including *gata-4/5/6, soxB1-2* and *FoxD*, are also expressed in planarian stem cells and are required for regeneration of the intestine (*Flores et al., 2016*; *González-Sastre et al., 2017*), sensory neurons (*Ross et al., 2018*), and anterior pole (*Scimone et al., 2014b*; *Vogg et al., 2014*), respectively. Therefore, upregulation of pioneer factors in stem cells may be a general strategy used to initiate organ regeneration. Identifying the regulatory mechanisms responsible for the selective activation of pioneer factors in stem cells may be an ideal approach to understanding how organisms initiate regeneration of targeted organs *in vivo*. In conclusion, our work sheds light on the flexibility and dynamic responses of stem cells to different injuries, and highlights potential mechanisms to activate organ-specific transcriptional programs required for regeneration.

# Materials and methods

## Key resources table

| Reagent type (species) or resource | Designation | Source or reference | Identifiers | Additional information |
|---|---|---|---|---|
| Antibody | Anti-DIG-AP (sheep polyclonal) | Roche | Cat#11093274910, RRID:AB_514497 | in situ: 1:3000 |
| Antibody | Anti-DIG-POD (sheep polyclonal) | Roche | Cat# 11207733910, RRID:AB_514500 | in situ: 1:1000 |
| Antibody | Anti-DIG_FITC (sheep polyclonal) | Roche | Cat# 11426346910, RRID:AB_840257 | in situ: 1:1000 |
| Antibody | Anti-phosphohistone H3 (Ser10) (rabbit monoclonal) | Abcam | Cat# Ab32107, RRID:AB_732930 | IF: 1:1000 |
| Antibody | Anti-Oregon Green-HRP (rabbit polyclonal) | Thermo Fisher | Cat# A21253, RRID:AB_2535819 | IF: 1:1000 |
| Antibody | Anti-tubulin (mouse monoclonal) | Sigma/Millipore | Cat# T5168, RRID:AB_477579 | WB: 1:1000 |
| Antibody | Anti-Phospho-p44/42 MAPK (Erk1/2) (rabbit monoclonal) | Cell Signaling Technologies | Cat# 4370S, RRID:AB_2315112 | WB: 1:1000 |
| Antibody | Goat anti-mouse Alexa Flour 488 (polyclonal) | Thermo Fisher | Cat# A11029, RRID:AB_2534088 | WB: 1:4000 |
| Antibody | Goat anti-rabbit IRDye 800CW (polyclonal) | LI-COR | Cat# 926–32211, RRID:AB_621843 | WB: 1:20,000 |
| Chemical compound, drug | F-ara-EdU | Sigma | Cat# T511293 | dilution: 0.5 mg/mL |
| Chemical compound, drug | Oregon Green 488 azide | Thermo Fisher | Cat# O10180 | F-ara-EdU development: 100 µM |
| Chemical compound, drug | Proteinase K | Thermo Fisher | Cat# 25530049 | F-ara-EdU development: 10 µg/mL in situ: 4 µg/mL |
| Chemical compound, drug | Roche Western Blocking Reagent | Roche | Cat# 11921673001 | dilution: 0.5% |
| Chemical compound, drug | Horse serum | Sigma | Cat# H1138-500mL | dilution: 5% |
| Chemical compound, drug | Nocodazole | Sigma | Cat# M1404 | dilution: 50 ng/mL |

*Continued on next page*

*Continued*

| Reagent type (species) or resource | Designation | Source or reference | Identifiers | Additional information |
|---|---|---|---|---|
| Chemical compound, drug | PD0325901 | EMD Millipore/Calbiochem | Cat# 4449685 MG | dilution: 10 µM |
| Chemical compound, drug | UO126 | Cell Signaling Technologies | Cat# 9903S | dilution: 25 µM |
| Chemical compound, drug | Western blot lysis buffer | *Zanin et al., 2011* | PMID:22118282 | |
| Chemical compound, drug | Pierce Protease Inhibitor | Thermo Fisher | Cat# A32965 | |
| Chemical compound, drug | Pierce Phosphatase Inhibitor | Thermo Fisher | Cat# A32957 | |
| Chemical compound, drug | Bolt LDS sample buffer | Life Technologies | Cat# B0007 | |
| Chemical compound, drug | Bolt 4–12% Bis-Tris polyacrylamide gel | Invitrogen | Cat# NW04125BOX | |
| Chemical compound, drug | Odyssey blocking buffer | LI-COR | Cat# 927–40000 | |
| Software, algorithm | GraphPad Prism | GraphPad Prism (https://graphpad.com) | RRID:SCR_002798 | Version 9 |
| Software, algorithm | GraphPad QuickCalcs | GraphPad QuickCalcs (https://graphpad.com/quickcalcs/) | RRID:SCR_000306 | |
| Software, algorithm | ImageJ | Image J https://imagej.net/ | RRID:SCR_003070 | |
| Other | DAPI stain 5 µg/mL | Thermo Fisher | | dilution: 1:5000 |
| Other | Aqua-Polymount | Polysciences Inc | Cat# 18606 | |
| Other | PVDF Immobilon membrane | Merck Millipore | Cat# IPFL00010 | |

## Worm care

Animals of *Schmidtea mediterranea* asexual clonal line CIW4 were maintained in a recirculating water system (*Arnold et al., 2016*; *Merryman et al., 2018*) containing Montjuïc salts (planaria water) (*Cebrià and Newmark, 2005*). Prior to experiments, animals were transferred to static culture and maintained in planaria water supplemented with 50 µg/mL gentamicin sulfate. Animals used for experiments were between 2 and 3 mm in length and starved for approximately 5–7 days.

## Amputations, sodium azide treatment, and tricaine anesthetization

Pharynx removal was performed by chemical amputation as previously described (*Adler et al., 2014*; *Shiroor et al., 2018*). Planarians (2–3 mm in size) were placed in 100 mM sodium azide diluted in planaria water. After 4–7 min, the pharynx extended out of the body and was plucked off using fine forceps (#72700-D; Electron Microscopy Sciences). Animals were kept in sodium azide for no longer than 10 min, rinsed three times, and then transferred into a fresh dish. For pharynx incisions and partial amputations, animals were soaked in tricaine solution (4 g/L in 21 mM Tris pH 7.5) diluted 1:3 in planaria water which causes the pharynx to extend but not detach. Pharynx incisions were created by using forceps to snip along the length of the pharynx. For partial pharynx amputations, the proximal end of the pharynx was snipped off with forceps or trimmed with a scalpel. To resect eyes, animals were immobilized on moist filter paper, and eyes were scraped out using the tips of fine forceps. All other amputations and injuries were performed with a micro feather scalpel (#72046–15 or #72045–45; Electron Microscopy Sciences). For direct comparisons to pharynx-amputated animals, head-amputated and intact animals were soaked in sodium azide for 2–3 min.

## F-ara-EdU administration

Animals were soaked in 0.5 mg/mL F-ara-EdU (Sigma T511293) in planaria water containing 3% DMSO for 4 hr either immediately or 24 hr after amputation and fixed 7 days after amputation.

### *In situ* hybridizations and immunostaining

Animals were fixed as previously described (*Pearson et al., 2009*) with minor modifications. Briefly, animals were killed in 7.5% N-acetyl-cysteine in PBS for 7.5 min and fixed in 4% paraformaldehyde in PBSTx (PBS + 0.3% Triton X-100) for 30 min. Worms were then rinsed twice with PBSTx and incubated in pre-warmed reduction solution (PBS + 1% NP-40 + 50 mM DTT + 0.5% SDS) at 37°C for 10 min. Worms were rinsed twice more with PBSTx, dehydrated in a methanol series and stored at −20°C.

For F-ara-EdU detection, following fixation, animals were rehydrated and bleached in 6% $H_2O_2$ overnight. Animals were then treated with proteinase K (10 µg/mL proteinase K and 0.1% SDS in PBSTx) for 15 min, and post-fixed in 4% formaldehyde in PBSTx for 10 min. A F-ara-EdU development solution was made containing PBS + 1 mM $CuSO_4$ and 100 µM Oregon Green 488 azide (Thermo Fisher O10180). Freshly made 100 mM ascorbic acid was added to this solution immediately before administering it to samples, which were then incubated for 30 min in the dark. Following a few rinses with PBSTx, animals were post-fixed, rinsed 2x in PBSTx, and put through *in situ* (see below). Following *in situ*, animals were placed in K block (5% inactivated horse serum, 0.45% fish gelatin, 0.3% Triton-X and 0.05% Tween-20 diluted in PBS) at room temperature for 4 hr or 4°C overnight. To detect F-ara-EdU, animals were incubated with 1:1000 anti-Oregon Green-HRP (Thermo Fisher A21253) and counterstained with DAPI in K block at 4°C overnight. Antibodies were washed off in PBSTx, pre-incubated with tyramide (1:2000 FAM) for 10 min and developed for 15 min.

Colorimetric *in situ* hybridizations were performed as described in *Pearson et al., 2009* using anti-DIG-AP (Roche 11093274910) at 1:3000. Fluorescent *in situ* hybridizations were performed as in *King and Newmark, 2013* with minor modifications. Briefly, animals were rehydrated and bleached (5% formamide, 1.2% $H_2O_2$ in 0.5x SSC) for 2 hr, then treated with proteinase K (4 µg/mL in PBSTx, Thermo Fisher 25530049). Following overnight hybridizations at 56°C, samples were washed 2x each in wash hybe (5 min), 1:1 wash hyb:2X SSC-0.1% Tween 20 (10 min), and 2X SSC-0.1% Tween 20 (30 min), 0.2X SSC-0.1% Tween 20 (30 min) at 56°C followed by 3 × 10 min PBSTx washes at room temperature. Subsequently, animals were placed in blocking solution (0.5% Roche Western Blocking Reagent and 5% inactivated horse serum diluted in PBSTx). Animals were then incubated with an appropriate antibody: 1:1000 anti-DIG-POD (Roche 11207733910) or 1:1000 anti-FITC-POD (Roche 11426346910) in blocking solution at 4°C overnight followed by several washes with PBSTx. For development with FAM (1:2000) or Cy3 (1:7500), animals were preincubated with tyramide in borate buffer for 30 min and then developed with 0.005% $H_2O_2$ in borate buffer for 45 min. For development with rhodamine, animals were pre-incubated with tyramide (1:5000) for 10 min and developed for 15 min. To inactivate peroxidases, animals were treated with 200 mM sodium azide or 4% $H_2O_2$ in PBSTx for 1 hr, then rinsed with PBSTx >6 times before application of the next antibody.

For H3P detection, following *in situ*, animals were incubated with anti-phosphohistone H3 (Ser10) antibody (Abcam, Cambridge, MA Ab32107) diluted 1:1000 in blocking solution (0.5% Roche Western Blocking Reagent and 5% inactivated horse serum in PBSTx) for 2 days at 4°C. Primary was washed off with PBSTx followed by incubation with goat anti-rabbit-HRP (Thermo Fisher 31460) diluted 1:2000 in PBSTx overnight at 4°C. Antibody was washed off with PBSTx and samples were pre-incubated and developed with rhodamine tyramide as described above.

For all *in situ* and immunostaining experiments, DAPI [5 µg/mL] (1:5000 dilution; Thermo Scientific) was added along with the last antibody (except for colorimetric *in situ*). After the final development, animals were soaked in ScaleA2 (4M urea, 20% glycerol, 0.1% Triton X-100, 2.5% DABCO) (*Hama et al., 2011*) for at least 3 days. Animals were mounted ventral side up except for those stained for *ovo*, which were dorsal side up, and embedded in Aqua-Polymount (Polysciences Inc 18606). To maintain consistent sample thickness, animals were mounted in wells cut from a double layer of double stick tape (Scor-Pal 6' wide Scor-Tape 209).

### Western blot

Ten animals per condition were snap frozen in lysis buffer (50 mM Hepes pH 7.5, 1 mM EGTA, 1 mM $MgCl_2$, 100 mM KCl, 10% glycerol, 0.05% NP40, and 0.5 mM DTT) (*Zanin et al., 2011*) containing Pierce protease and phosphatase inhibitors (Thermo Fisher A32965 and A32957). A cup horn sonicator (Branson Ultrasonics Corporation, Danbury, CT) chilled to 4°C was used to generate extracts by sonication for a total of 2 min with 1 s pulses at 90% amplitude. Total protein was quantified using a

NanoDrop One$^C$ (Thermo Fisher). After quantification, Bolt LDS sample buffer (Life Technologies B0007) was added to the extracts and 100 µg of each sample was run on a polyacrylamide gel (Bolt 4–12% Bis-Tris, Invitrogen NW04125BOX). The gel was transferred onto a PVDF Immobilon membrane (Merck Millipore IPFL00010) using the Pierce Power Blot Cassette system (Thermo Scientific), then treated with Odyssey blocking buffer (LI-COR 927–40000) for 1 hr at RT. Membranes were incubated overnight at 4˚C with mouse anti-tubulin (Sigma/Millipore T5168) and rabbit anti-Phospho-p44/42 MAPK (Erk1/2) (Cell Signaling Technologies 4370S), diluted 1:1000 in blocking buffer. The blot was washed 3 × 10 min in TBST (TBS +10% Tween 20) and incubated for 1 hr at RT with goat anti-mouse Alexa Flour 488 (Thermo Fisher A11029) and goat anti-rabbit IRDye 800CW (LI-COR 926–32211) secondary antibodies (diluted 1:4000 and 1:20,000 respectively in Odyssey blocking buffer). Membranes were then washed as before and imaged using a Bio-Rad ChemiDoc MP. Western blots were repeated at least twice with comparable results.

## Drug treatments

Nocodazole (Sigma M1404) was administered in 24 or 48 hr increments at 50 ng/mL. PD0325901 (EMD Millipore/Calbiochem 4449685 MG) and UO126 (Cell Signaling Technologies 9903S) were administered at 10 µM and 25 µM, respectively. Drugs were diluted in planaria water containing 0.05% DMSO. Animals were rinsed three times after treatment and either fixed immediately, or transferred to a new dish and rinsed daily until further experimentation.

## Feeding assay

Animals were fed 20 µL of colored food (4:1 liver:milliQ water with 2% red food coloring) in a petri dish. After approximately 30 min, the number of animals with red intestines were scored. For time courses, feeding assays started at 4 days post-amputation and any animals that ate were removed from the dish. Feeding assay time courses were repeated at least three times with ~20 animals assayed per experiment.

## RNA interference

RNAi was performed as previously described (*Rouhana et al., 2013*), with *in vitro*-synthesized double-stranded RNA (dsRNA). dsRNA was diluted to a final concentration of 400 ng/µL in colored food. RNAi food was administered every 3 days, six times in total, except for *gata-4/5/6* and *six-1/2*, which caused phenotypes after 1–2 feeds. *C. elegans unc22* dsRNA was used as a control. Amputations were carried out 5–7 days after the last feed. All RNAi experiments were repeated at least twice with ~10 animals per experimental group.

## Image acquisition, quantification, and statistical analysis

Whole-mount colorimetric *in situ* hybridizations and live worms were imaged on a Leica M165F. Fluorescent *in situ* hybridizations were imaged on a Zeiss 710 confocal microscope using a 25x objective with 2.28 µm z-sections. ImageJ software was used for processing and quantification (*Schindelin et al., 2012*). All samples were quantified without blinding by manual examination of optical sections of overlaid fluorescence channels in pre-defined regions of animals as indicated in figures. Cells were identified as positive for markers if fluorescence coincided with DAPI signal and was easily distinguishable from background levels, as demonstrated in figures with images. Quantification was performed in a minimum of 6 animals per experimental group.

For *piwi-1*$^+$ progenitor analysis, a 6000 µm$^2$ region in the same location of the pre-pharyngeal area was captured at 1.3x zoom. Quantification included 20 z-sections (45.6 µm) beginning at the first *piwi-1*$^+$ cell. For H3P analysis, the entire pre-pharyngeal region was imaged, captured at 0.6x zoom. Quantification included 30 z-sections (68.4 µm) beginning at the first H3P$^+$ cell and was normalized to area. Representative confocal images are partial projections of ~5 z-sections from regions that were used for quantification, re-imaged at 4x zoom. For F-ara-EdU quantification, images were captured at 1x zoom. In most cases, the pharynges and brains of each experimental group were imaged from the same animal. All visible F-ara-EdU$^+$ cells in the pharynx and all F-ara-EdU$^+$ *ChAT*$^+$ cells in the brain were quantified throughout the entire organ. Representative F-ara-EdU images are projections of the entire analyzed region. In bar graphs, symbols represent individual animals, and shapes distinguish biological replicates. Statistical analysis was performed using PRISM-Graphpad

version nine or GraphPad Quick Calcs to perform one-way ANOVA with Tukey test, unpaired t-test or Fisher's Exact test as indicated in figure legends. *$p \leq 0.05$; **$p \leq 0.01$; ***$p \leq 0.001$; and ****$p \leq 0.0001$.

## Primers

Sequences for all transcripts used in this study were cloned using the following primers:

| Gene | Smed ID | Forward primer | Reverse primer |
|---|---|---|---|
| piwi-1 | dd_Smed_v6_659_0_1 | gaccaagaagaggaggtctcc | gcgttcgcgaattctgtcatt |
| FoxA | dd_Smed_v6_10718_0_4 | aacgacctcaacggaatgttt | catgcgccaaagttaaggata |
| ovo | dd_Smed_v6_48430_0_1 | aatgcccacagatttgtc | cataaagtgaattcgggtg |
| myoD | dd_Smed_v6_12634_0_1 | ctattccggtccatactcagc | actcttgatcaactttcctcg |
| gata-4/5/6 | dd_Smed_v6_4075_0_1 | gtccgtaagatccacgatccg | tgattgaggaatagggcttcg |
| six-1/2 | dd_Smed_v6_9774_0_1 | ccttgtcagggatctaatcc | ggtgaggatgataagttggg |
| pax6a | dd_Smed_v6_17726_0_1 | ctgggcataaatcaaaccgc | cttgggggataaactgatcc |
| coe | dd_Smed_v6_9893_0_1 | cgaagagcagacaacagcac | ttttaccaacacccgattgc |
| laminin | dd_Smed_v6_8356_0_1 | agtcgctggcaaagtgcatct | aatgatgcgtggtatccacag |
| fst | dd_Smed_v6_9584_0_1 | cagtggtgtgcaatttagcgagttc | gcaggtattcttggtttcgtaattcg |
| ChAT | dd_Smed_v6_6208_0_1 | tcggttgctgaaggtattgca | ggcatatagcattctacacgg |

## Acknowledgements

We thank the Cornell University Biotechnology Resource Center for assistance with data collection on the Zeiss LSM 710 Confocal which is supported by the NIH (NIH S10RR025502). We would also like to thank Kuang-Tse Wang and Justin Tapper for helpful comments on the manuscript and Melanie Issigonis for assistance with F-ara-EdU-labeled FISH. DAS was supported by a GRA Fellowship from the Cornell College of Veterinary Medicine. This work was supported by Cornell University startup funds and R01GM139933 awarded to CEA.

## Additional information

### Funding

| Funder | Grant reference number | Author |
|---|---|---|
| National Institute of General Medical Sciences | R01GM139933 | Carolyn E Adler |
| College of Veterinary Medicine, Cornell University | | Divya A Shiroor |

The funders had no role in study design, data collection and interpretation, or the decision to submit the work for publication.

### Author contributions

Tisha E Bohr, Conceptualization, Data curation, Formal analysis, Validation, Investigation, Visualization, Methodology, Writing - original draft, Writing - review and editing; Divya A Shiroor, Conceptualization, Data curation, Formal analysis, Investigation, Visualization, Methodology, Writing - original draft, Writing - review and editing; Carolyn E Adler, Conceptualization, Supervision, Funding acquisition, Writing - original draft, Project administration, Writing - review and editing

## Author ORCIDs

Tisha E Bohr (iD) https://orcid.org/0000-0003-2336-407X
Divya A Shiroor (iD) https://orcid.org/0000-0001-9085-5675
Carolyn E Adler (iD) https://orcid.org/0000-0002-3883-0654

## Decision letter and Author response

Decision letter https://doi.org/10.7554/eLife.68830.sa1
Author response https://doi.org/10.7554/eLife.68830.sa2

## Additional files

### Supplementary files

• Supplementary file 1. Table of primers and plasmids.

• Transparent reporting form

### Data availability

All data generated and analyzed in this study are included in the manuscript and supporting files. Numerical data used to generate all graphs are included in a single Source Data File with an individual tab containing the raw data for each figure panel.

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
