## [Decision Letter]

**Acceptance summary:**

How stem cells respond to missing tissue is an important question for understanding regenerative processes. Here, Bohr et al. address this question by comparing how stem cells respond to loss of a single organ (the pharynx) versus loss of many tissues (after head amputation) in the planarian, an organism that can regenerate its entire body from a tiny piece of tissue. The authors find that the stem cells respond to loss of the pharynx by producing more pharynx progenitors; this increase is not observed after removal of non-pharyngeal tissues. Thus, the planarian's stem cells are able to "sense" when certain tissues are missing and target their fates accordingly.

**Decision letter after peer review:**

[Editors’ note: the authors submitted for reconsideration following the decision after peer review. What follows is the decision letter after the first round of review.]

Thank you for submitting your work entitled "Planarian stem cells sense the identity of missing tissues to launch targeted regeneration" for consideration by *eLife*. Your article has been reviewed by 3 peer reviewers, one of whom is a member of our Board of Reviewing Editors, and the evaluation has been overseen by a Senior Editor. The reviewers have opted to remain anonymous.

Our decision has been reached after consultation between the reviewers. Based on these discussions and the individual reviews below, we regret to inform you that your work will not be considered further for publication in *eLife*.

All three reviewers recognized the potential significance of this work, but also shared the same concerns about sample sizes, lack of biological replicates, and insufficient details about cell quantification. Given the interest in the question being pursued, if these issues can be addressed satisfactorily, a revised paper could be considered here as a new submission. We have included the reviews below and hope you will find the comments helpful.

*Reviewer #1:*

This manuscript by Bohr et al. explores how planarian stem cells respond to the loss of a specific organ: the pharynx. The previously proposed "target-blind" model of planarian regeneration (LoCascio et al. 2017) posited that stem cells do not respond directly to missing tissues, but rather replace missing cell types based on their normal rates of homeostatic turnover. In contrast, Bohr et al. suggest that planarian stem cells can sense and respond to the loss of specific missing tissues, using the pharynx as a case study. The authors conclude that planarians may use more than one mode of regeneration, depending upon the target being regenerated (eye vs. pharynx).

The question explored in this paper is of fundamental importance, and providing an alternative model by which planarian stem cells regenerate missing tissues should be of interest to a broad readership. Unfortunately, in its current form, the manuscript presents enticing preliminary findings, rather than robust experimental observations. Currently, the manuscript has limited samples sizes and experimental replicates, which is unfortunate. Because this paper is attempting to refute a previously published model it is critical that the data are clear and convincing. If not, these findings could be summarily dismissed without appropriate debate. If the authors can show the robustness and rigor of their results, and address the major issues listed below, this manuscript would represent a significant contribution to our understanding of planarian regeneration.

1. Throughout the manuscript, experiments were either not repeated, or the number of biological replicates was not reported. In most cases, it appears that experiments were done only once (with the exception of the drug treatments). Numbers of biological replicates and sample sizes should be explicitly stated and the data from different replicates reported for Figures 1D-G, 2B-D, 3C, 3E-F, 4B, 4D-H, 5C-D, and 6A-E.

2. The authors do not sufficiently describe their methods for imaging and quantifying cells (Figures 1E, 1G, 2C-D, 3F, 4E-H, 5D, 6B). The size of the area covered to collect these data is unclear. High-magnification images are shown: are these the areas that were imaged? If so, their results could be biased by choosing small regions of interest. Ideally, the authors should quantify more than one region per animal. Also, they do not describe the depth of the z-stacks collected or how these stacks were normalized/standardized across conditions. All their conclusions hinge on the quantification of progenitor populations in response to different amputation paradigms or chemical treatments, so the standards for imaging and quantification must be clearly reported.

3. Inappropriate statistical tests were used throughout. The use of multiple t-tests amplifies the chance of a Type I error and is especially problematic when up to 7 comparisons were made! The authors should use one-way ANOVA with multiple comparison corrections for all experiments with more than two groups.

4. Figures 1D-E show that upon pharynx amputation but not head amputation, FoxA+ piwi+ pharynx progenitors increase. These data suffer from the quantification issues highlighted above: how the data were quantified is not sufficiently described, only 3 data points were taken (one per animal), the experiment appears to have been performed only one time, and the wrong statistical test was used. Rather than reporting the number of FoxA+ piwi-1+ cells counted, the authors should quantify the total number of doubly positive cells as a percentage of piwi-1+ cells, as was previously published (Adler et al. 2014). The authors also fail to specify whether the change observed between "3 dpa phx" and "3 dpa head" is significant, which is a material point.

5. Figure 2D also suffers from the inadequate quantification practices described above. Ideally, FoxA+ cells should be quantified as a percentage of the H3P+ cells observed.

6. The authors use "stem cell", "progenitor", "stem cell progenitor", and "progenitor stem cells" in a mixed and confusing way throughout the paper. For example, in lines 174-175 the authors state that "proliferation of FoxA+ stem cells precedes the increase in pharynx progenitors." This refers to FoxA+ H3P+ cells vs. FoxA+ piwi-1+ cells, but the only difference is that the former are stem cells in the act of mitosis. Is a distinction being made? Elsewhere in the paper, FoxA+ piwi+ cells are referred to as stem cells. The terminology used needs more clarity and consistency.

Along these lines, what is a FoxA+ PSC (Line 168)? Are the authors suggesting that FoxA is a pluripotency marker? Or are the authors saying that tgs^-1^ has a more expansive expression pattern and is co-expressed with progenitor markers? If double FISH was performed with the progenitor markers reported (ovo, myoD, gata-4/5/6, six-1/2, and pax6), would they all overlap with tgs^-1^? These experiments need to be performed to make any claims about FoxA expression in the context of pluripotency.

7. In Figure 3C, how does pharynx regeneration occur over such a long period of time after nocodazole treatment in the 1-2 day window? Does target-blind regeneration occur once this window is missed? The authors should repeat analysis of FoxA+ progenitors at later time points in this condition, and/or show rates of BrdU incorporation into the pharynx with and without nocodazole treatment in this window.

8. The use of the inhibitor PD in Figure 4 is problematic. No data are shown to verify that phosphorylation of ERK was inhibited in these experiments. A citation of previous use is not sufficient. The effect of PD on WT uncut animals with regards to FoxA+ cells is not shown and is a necessary control. To address questions of drug specificity, the authors should corroborate their findings with a second inhibitor of ERK signaling like U0126, which has already been shown to work in planarians (Owlarn et al. 2017).

9. The conclusion that ERK signaling functions in regulating differentiation but not proliferation is premature (line 249-264). Figures 4E-H should be quantified as percentages of piwi-1+ and H3P+ cells, especially since there is decreased proliferation overall with PD treatment.

10. The use of head fragments to compare eye regeneration vs. pharynx regeneration is inappropriate. Previous studies have already shown that the absence of eyes is not required to induce ovo+ progenitor amplification (LoCascio et al. 2017). Thus, this is not a surprising result (line 343-345) and the authors are mis-citing previous observations (in the earlier work, head fragments were never described). The region where ovo+ cells was quantified in Figure 6A is not justified or explained. The yellow box is placed in a medial region where ovo+ cells do not normally reside. The authors should image within the laterally positioned ovo+ streams that have been previously described (Lapan and Reddien 2012; LoCascio et al. 2017).

11. Rather than using head fragments, the authors should repeat the flank resection experiments shown previously (LoCascio et al. 2017). This previous study showed that increased BrdU incorporation into the pharynx occurred following flank resection even though the pharynx was present. That result may have been 1) an artifact of increased BrdU staining due to stimulation of proliferation upon injury, 2) caused by unintentional damage to cells associated with the pharynx, or 3) a response to the loss of FoxA+ progenitor populations that surround the pharynx rather than the loss of the differentiated organ. The authors have the opportunity to revisit this published observation by quantifying the FoxA+ progenitor response during flank resection +/- pharynx. Without these data, this story is incomplete and therefore the conclusion of a targeted regeneration response is not yet convincing.

12. The negative results that proliferation and ERK signaling are dispensable for eye regeneration in Figure 6 are weak and unconvincing. The regenerated eyes appear smaller; this should be quantified (number of PRNs per eye). If the small pharynges that form in Figures 3D and 4C are considered a deleterious phenotype, why is the same standard not applied to the eye? Also, existing eye progenitors could have been sufficient for eye regeneration in these drug treatments. Furthermore, eyes did not regenerate after nocodazole treatment 50% of the time. Is it not more likely that the observations reported are dosage and timing artifacts? How has proliferation been affected? These observations do not live up to the claims made.

13. The authors claim that ovo+ cells are not proliferative (H3P+) even in cases where there is eye progenitor amplification (head amputation), but the data are not shown (line 321). They should be. Indeed, previous publications have never shown that ovo+ cells proliferate. This might mean that there are proliferating eye progenitors that precede expression of ovo. The authors should discuss this alternative.

14. The authors claim that eye regeneration does not require proliferation or ERK signaling but pharynx regeneration does. This conclusion hinges on the gross observation that eyes can regenerate in the presence of nocodazole and PD (see point 12 above). These data are coarse and the interpretations are unconvincing. Instead, their model can be directly tested using BrdU pulse-chase experiments. According to the authors' model, one would predict that following pharynx amputation, the rate of incorporation of BrdU^+^ cells into the regenerating pharynx should be higher than in uninjured controls. Conversely, the rate of BrdU incorporation into the regenerating eye should remain unchanged between injured (i.e. eye-resected) and control animals (LoCascio et al. 2017). Once the authors have established the prediction above, they have the opportunity to show the effects of nocodazole, PD, and U0126 on BrdU incorporation in the regenerating eye vs pharynx following eye resection and pharynx amputation in the same animal. This way, the authors can directly test the requirement for proliferation and/or ERK signaling in both tissues.

*Reviewer #2:*

Bohr, Shiroor, and Adler investigate how stem cells respond to the loss of specific tissues in planarians. The planarian stem cell population (neoblasts) are distributed throughout the planarian body and include pluripotent stem cells and a wide range of lineage-committed progenitor cells. How this heterogenous pool of cells behave post-injury or amputation is incompletely understood. The discovery of markers to label stem cell progeny have opened the door to investigate stem cells respond to tissue loss. However, the anatomy of planarians makes it difficult to surgically remove or damage specific organs. The PI of this study developed an assay to remove the pharynx by "chemical amputation" to study the mechanisms underlying regeneration of this organ without drastically perturbing or injuring other tissues. Using this approach, this paper investigates how a well-defined population of FoxA+ progenitors respond to pharynx removal at early time-points during the regeneration. Their data suggest that stem cells are able to detect loss of the pharynx and respond by generating significantly more cells fated to become pharynx, whereas amputation of non-pharyngeal tissues does not have an obvious effect on pharynx progenitor specification dynamics. In addition, using pharmacological treatments, the authors show that cell proliferation and ERK signaling are required for the expansion of pharynx progenitors and cell differentiation. In contrast, other cell types in the planarian eye do not appear to require proliferation or ERK signaling, suggesting that stem cell responses "target blind" as suggested in a previous study, but are rather tuned to specific missing tissues.

This work has the potential to make a significant contribution to the field by advancing our understanding of how the planarian heterogenous stem cell population responds to the loss of a specific organ. However, the report is preliminary as presented. It appears that the authors performed many experiments a single time. In addition, description of the methods is insufficient. Consequently, before this work can be considered for publication, the authors need to chiefly demonstrate the reproducibility of the data and robustness of the observations.

1. The authors need to replicate experiments to increase the sample size for most experiments.

2. Details for imaging and quantification should be explicitly stated in the methods, and the reported cell count numbers should be normalized as appropriate for each set of experiments.

3. Although the authors mention "lineage-tracing" experiments, they do not perform DNA analog pulse-chase experiments to analyze a temporal progression and spatial localization of stem cells to FoxA+ progenitors after pharynx removal. The authors rely on PH3-staining in conjunction with FoxA, and supplementary experiments using the pluripotent stem cell marker tgs^-1^ (which was only examined at 1 dpa). Could the authors clarify what they think FoxA+ stem cells represent? Are these self-renewing pluripotent stem cells or lineage-committed progenitors? Can the authors get some insight by scanning their images of PH3+ cells expressing FoxA visibly undergoing metaphase? Are daughter cells uniformly FoxA+ as reflected in the model? At least in one of the cells shown in the nocodazole treated controls it appears that both daughter cells express FoxA (Figure 3). I suggest showing some higher magnification images to support the interpretations/conclusions. Others have posited (e.g., Rink, Chapter 2 of Planarian Regeneration: Methods and Protocols), that not every dividing cell may be a long-term self-renewing stem cell and whether a transient amplifying cells exist or contribute to regeneration in planarians is unknown. Adler and Sánchez Alvarado (2015) discuss the role of transient states and how the transcriptional profiles change in response to regeneration. It wasn't clear to me how the authors think about these cells based on the limited number of experiments and analysis, and there are a few places where the terminology is inconsistent, especially in reference to proliferating ovo+ progenitors (P. 14). The authors need to be clear, and it might be helpful to illustrate their model in one of the early figures or to include it in the final model, which omits tgs^-1^ due to the limited number of experiments performed with this marker gene (Figure 7).

4. The pharynx is complex and there is no data to assess what the contribution of other progenitor populations might be. I don't think or think it is unlikely that FoxA+ progenitors are solely responsible for reconstructing the pharynx. The authors should examine how other progenitor populations behave during the process of pharynx regeneration by extending the timeline of progenitor cell analysis. This would reveal if there is fluctuation in progenitor dynamics as animals regenerate the pharynx or re-scale proportions after pharynx regeneration. For example, can the authors test if they are able to detect a contribution of neural progenitors to regeneration of the pharyngeal nervous system? And if so, when during the regeneration process does it take place in the context of their study?

*Reviewer #3:*

In this manuscript, Bohr et al. examine how the pluripotent stem cell system of planarians responds to organ-specific damage. If and how the differentiation of specific cell types is dynamically regulated is a conceptually fascinating problem in planarians and in general stem cell research. The authors address this problem by comparing the stem cell response between a single-organ amputation (the pharynx) versus broad tissue loss (decapitation). Their findings indicate that only the removal of the pharynx triggers the increased differentiation of pharyngeal cell types, while the loss of non-pharyngeal tissues upregulates the differentiation of progenitors of multiple organs, but not the pharynx. Further, the authors implicate temporally restricted ERK signaling as a regulatory component in the differentiation of pharyngeal cell types. These observations are also important because they contrast with the previously proposed "target blind" model (LoCascio et al., 2017) that posits the differentiation of different cell types at constant relative proportions, with the rate of stem cell divisions as global production rate regulator. In contrast, the observations by Bohr et al. provide further evidence for more flexibility and specificity within the planarian stem cell system ("target consciousness") in the sense of lineage-specific adjustments in the production rates of specific cell types.

That said, the manuscript generally suffers from an overly narrow focus. Important questions remain regarding the specificity of the stem cell response to pharynx amputation and multiple experiments lack important controls (see below). Moreover, the authors have overlooked that a "target conscious" progenitor response has already been demonstrated by the selective proliferation of protonephridial marker expressing neoblasts in response to protonephridial damage by RNAi (Vu et al., 2015). As a result, the current manuscript would require substantial additional experimentation to consolidate its findings sufficiently for publication in *eLife*.

1. Specificity of the stem cell response

The central premise of the paper is the selective amplification of pharyngeal progenitors in response to pharynx amputation. This the authors conclude on basis of i) an increase in the absolute number of foxA+/piwi-1+ cells in a specific area, while ii) de-capitation has no effect on the absolute number of foxA+/piwi-1+ cells in the same area. This approach is an insufficient demonstration of specificity, as the known phenomenon of wound-induced stem cell activation might also change the absolute number of specific neoblast subclasses and might do so in an injury-dependent manner.

To account for this important caveat, the authors need to i) quantify RELATIVE proportions of foxA+/piwi-1+ cells out of total piwi cells (or of total H3P+ cells) and ii) they need to include other organ progenitors in the initial analysis. The latter is also critical because the pharynx is a complex organ comprising descendants of multiple lineages (e.g., muscle, neurons, epidermis) and it is not clear whether the foxA+/piwi-1+ cells indeed serve as a singular origin of all constituting lineages (as assumed by the authors), or if they only provide a subset of pharyngeal cells with a rate-limiting role in pharynx assembly (e.g., pharyngeal muscle). In face of such uncertainty, iii) the quantification of new cell incorporation into the pharynx versus other tissues via BrdU labeling would be necessary to address this caveat and to provide a global perspective on the specificity of the response independent of incompletely characterized marker genes.

In addition, the following experimental design problems need to be addressed or better documented, including:

• The authors provide insufficient methodological detail on progenitor quantifications, even though the entire manuscript rests on this assay. What are their criteria for scoring a piwi-1+ cell as double-positive for the often weak and noisy lineage labels? If done "by eye", was double-blind scoring used? Were all cells in a given Z-stack counted or only specific planes? If the latter, by which criteria were image planes selected for quantification? How were the specific specimens out of an experimental cohort selected for imaging/quantification? Though not necessarily a unique shortcoming of this particular study, these points simply need to be adequately addressed in order to rigorously support quantitative differences between experimental conditions (e.g., specificity).

• The authors appear not to distinguish at all between technical replicates (e.g., multiple specimens within an experimental cohort) versus biological replicates (independent experimental cohorts). This is significant, because (i), the use of the standard error of the mean (SEM) that the authors employ throughout is not really an appropriate measure for a single biological replicate with 3 animals – the standard deviation (SD) would seem a more appropriate measure in this context (SD). (ii), the number of worms quantified for each experiment is generally low (n=3 animals in Figure 1, 3, 5, 6; n=5 animals in the rest of the figures) given the observed variability in the data (e.g., ~25-30 foxA+/piwi-1+ cells 3d after pharynx amputation in Figure 1E versus 50 cells in figure 1H). Similarly, for kinetic experiments as in Figure 1H or 2C, it is simply crucial to ensure that the error bars include the variation in response dynamics between multiple replicates due to the drift in the baseline fraction of H3P+ cells or varying staining efficiencies (e.g., different batches of animals on different days), rather than the technical variation in a single experimental cohort only. Please address these concerns by adding more specimens and a thorough description of the experimental design.

2. Timeline of pharynx regeneration

The pharynx regeneration timeline and associated events that the authors present are insufficiently supported by experimental data. The conclusion in line 215 "that proliferation in a critical window of 1 to 2 days after pharynx amputation produces a population of progenitors that are likely essential for pharynx regeneration" rests (i) on the diagnosis of a "proliferative peak of FoxA+ stem cells that occurs after pharynx amputation (Figure 2C)" (line 202). However, rather than a "proliferative peak", Figure 2C shows a broad "proliferative plateau" of FoxA+ stem cells between 6h and 3 days after amputation. Similarly, the foxA+/H3P+ quantification after pharynx amputation in Figure 4G also displays a lack of a peak of foxA+/H3P+ from 1d to 2d after amputation. (Ii), the associated nocodazole experiments suffer from the fact that the authors did not quantify the impact of the drug on the abundance of foxA+/piwi-1+ cells during treatment intervals from 0-1d and 2-3d after pharynx amputation. Therefore, the authors cannot rule out that nocodazole treatment might have similar effects on the abundance of foxA+/piwi-1+ cells throughout the 1-3 d post-amputation time interval, with the more severe organ-level phenotype of the 1-2d treatment window being caused by some other effect of the drug (e.g., on the differentiation of another rate-limiting cell type for pharynx regeneration or, conceivably, inhibition of priming neuronal activity ). Similar concern apply regarding the statement in line 242, "… a window 1-2 days after amputation in which activation of ERK signaling is important for pharynx regeneration." Here, (i) the quantification of the end-stage phenotype of drug treatment during the 1-2d time interval (regain of feeding ability) is missing. (ii) Similarly, the examination of the consequences of PD treatment on foxA+ expression in piwi-1+ cells in panel 4D-H employs drug soaking for 3 days, yet the corresponding end-stage phenotype of 3-day drug treatment is not shown. (iii) the implications of ERK in pharynx regeneration are tentative. Even though the PD compound is initially correctly introduced as "MEK inhibitor", the authors subsequently switch to the factually wrong "ERK inhibitor" designation (e.g., line 358). Further, additional experimental evidence for the assumed Erk inhibition as the cause of the observed phenotypes would be desirable to rigorously support the conclusion.

These caveats need to be addressed if a cell biological timeline is to remain part of this manuscript.

3. Integration with the existing literature

The authors need to better integrate their findings with the literature. First, they need to cite the findings of Vu et al., which explicitly demonstrated a specific increase in the fractional abundance of piwi-1+/protonephridial marker+ cells in response to RNAi-mediated damage to protonephridia(Vu et al., 2015). As such, this study already demonstrates the main point of Bohr et al., namely that the planarian stem cell system is capable of "target conscious" progenitor provision. In the very least, the authors should credit these results as additional evidence for their model. A further finding that they should discuss is the demonstration by LoCascio et al. (LoCascio et al., 2017) that flank region cut-outs cause a significant increase in pharynx cell incorporation over baseline, despite the absence of injury to the pharynx. How do the authors reconcile the discrepancy between these data and their own? In general, the discussion would benefit greatly from a more explicit comparison between the "target blind" model versus their data, as well as a broader perspective on the regulation of stem cell homeostasis.

---

## [Author Response]

All three reviewers recognized the potential significance of this work, but also shared the same concerns about sample sizes, lack of biological replicates, and insufficient details about cell quantification. Given the interest in the question being pursued, if these issues can be addressed satisfactorily, a revised paper could be considered here as a new submission. We have included the reviews below and hope you will find the comments helpful.Reviewer #1:This manuscript by Bohr et al. explores how planarian stem cells respond to the loss of a specific organ: the pharynx. The previously proposed "target-blind" model of planarian regeneration (LoCascio et al. 2017) posited that stem cells do not respond directly to missing tissues, but rather replace missing cell types based on their normal rates of homeostatic turnover. In contrast, Bohr et al. suggest that planarian stem cells can sense and respond to the loss of specific missing tissues, using the pharynx as a case study. The authors conclude that planarians may use more than one mode of regeneration, depending upon the target being regenerated (eye vs. pharynx).The question explored in this paper is of fundamental importance, and providing an alternative model by which planarian stem cells regenerate missing tissues should be of interest to a broad readership. Unfortunately, in its current form, the manuscript presents enticing preliminary findings, rather than robust experimental observations. Currently, the manuscript has limited samples sizes and experimental replicates, which is unfortunate. Because this paper is attempting to refute a previously published model it is critical that the data are clear and convincing. If not, these findings could be summarily dismissed without appropriate debate. If the authors can show the robustness and rigor of their results, and address the major issues listed below, this manuscript would represent a significant contribution to our understanding of planarian regeneration.

We thank the reviewer for recognizing the importance of the questions we have pursued here. However, we would like to clarify that our intention was not to refute the previously published model. This confusion likely arose because of a lack of clarity in our writing. In the current version, we explicitly test the ‘non-targeted’ model with EdU incorporation (Figure 1E,F and Figure 1—figure supplement 1) and flank resection experiments (Figure 2D). Our data shows that the ‘targeted’ model is the primary mode by which the pharynx regenerates, but the ‘non-targeted’ model also operates immediately after wounding. We have added an entire section in the discussion to make it plainly apparent to readers that both mechanisms likely operate during regeneration.

1. Throughout the manuscript, experiments were either not repeated, or the number of biological replicates was not reported. In most cases, it appears that experiments were done only once (with the exception of the drug treatments). Numbers of biological replicates and sample sizes should be explicitly stated and the data from different replicates reported for Figures 1D-G, 2B-D, 3C, 3E-F, 4B, 4D-H, 5C-D, and 6A-E.

We agree that the manuscript suffered from a general lack of experimental replicates and small sample sizes, as addressed in the overview above. Taking into account these essential criticisms, we have repeated each experiment 2-3 times, increased overall sample sizes, and explicitly marked distinct biological and technical replicates on graphs, where possible. We have also included a table detailing biological and technical replicates for each relevant figure.

2. The authors do not sufficiently describe their methods for imaging and quantifying cells (Figures 1E, 1G, 2C-D, 3F, 4E-H, 5D, 6B). The size of the area covered to collect these data is unclear. High-magnification images are shown: are these the areas that were imaged? If so, their results could be biased by choosing small regions of interest. Ideally, the authors should quantify more than one region per animal. Also, they do not describe the depth of the z-stacks collected or how these stacks were normalized/standardized across conditions. All their conclusions hinge on the quantification of progenitor populations in response to different amputation paradigms or chemical treatments, so the standards for imaging and quantification must be clearly reported.

We agree that the previous version of the manuscript lacked clarity with regard to imaging and quantification. We have now addressed these important criticisms by clearly describing them in the figure legends and the Image acquisition, quantification and statistical analysis section of the methods (beginning on line 759). We now explicitly state the following standardized parameters for a given experiment: 1) the region analyzed within the animal, 2) the x-y area measured, 3) thickness (z-sections) for a given experiment, and 4) the z-section where quantification began.

3. Inappropriate statistical tests were used throughout. The use of multiple t-tests amplifies the chance of a Type I error and is especially problematic when up to 7 comparisons were made! The authors should use one-way ANOVA with multiple comparison corrections for all experiments with more than two groups.

As suggested, all figures in which multiple comparisons are made now use one-way ANOVA to determine statistical significance.

4. Figures 1D-E show that upon pharynx amputation but not head amputation, FoxA+ piwi+ pharynx progenitors increase. These data suffer from the quantification issues highlighted above: how the data were quantified is not sufficiently described, only 3 data points were taken (one per animal), the experiment appears to have been performed only one time, and the wrong statistical test was used. Rather than reporting the number of FoxA+ piwi-1+ cells counted, the authors should quantify the total number of doubly positive cells as a percentage of piwi-1+ cells, as was previously published (Adler et al. 2014). The authors also fail to specify whether the change observed between "3 dpa phx" and "3 dpa head" is significant, which is a material point.

We have addressed the concerns with biological replicates, sample sizes and statistical analysis as described in our responses to points 1 and 3. We have also attempted to indicate when relevant differences are not statistically significant, as is the case for the comparison in question above.

As for representing our data as the absolute number of cells rather than as a proportion, the reviewer is correct in noting that this strategy is a departure from our previous paper, where *FoxA*^+^*piwi-1*^+^ cells were represented as a relative percentage (Adler et al., 2014). However, similar quantification of ovo^+^ eye progenitors is represented as absolute numbers rather than percentages (Lapan et al. and LoCascio et al.). Without clear quantification standards in the literature, we initially evaluated both strategies, but found the results to be very similar. We have included two figures of side-by-side comparisons of these two quantification strategies for key panels from our manuscript (Author response images 1 and 2). Given the complex dynamics of the stem cell population, we feel that representing the data as relative percentages oversimplifies our analysis. Because our progenitor analysis initially compares multiple injury scenarios, we are confident that the changes we see in progenitors occur independently of broad injury responses. Further, representing the data as means of absolute numbers of cells in a standardized area/thickness allows us to indicate individual animals and biological replicates on graphs, providing a more transparent view of the data to readers. Based on these criteria, we have maintained our strategy of representing the data as absolute values, along with extensive additions to the manuscript clarifying where and how the data were obtained.

**Author response image 1. sa2fig1:** Pharynx loss selectively increases pharynx progenitors in proportion to stem cells. (A) Proportion of cells double positive for the indicated progenitor marker and *piwi-1^+^* relative to all *piwi-1^+^* stem cells in the area outlined by dashed boxes in cartoons. Cartoons depict different amputation conditions. n ≥ 790 cells per experimental group from 3 independent experiments. (B) Average number of *FoxA^+^ piwi-1^+^* cells in the same animals and regions as A. Same data as is in Figure 2B, E of manuscript. (C) Proportion of *FoxA^+^ piwi-1^+^* cells at indicated times post-pharynx amputation relative to all *piwi-1^+^* stem cells in the area outlined by dashed boxes in A. n ≥ 631 cells per experimental group from 3 independent experiments. (D) Average number of *FoxA^+^ piwi-1^+^* cells in the same animals and regions analyzed as C. Same data as in Figure 2C of manuscript. For all graphs a 6000μm^2^ region in the same location of the pre-pharyngeal region was analyzed over 20 z-sections, as represented by dashed boxes in A. Graphs represent a proportion ± 95% confidence intervals (A, C) or the mean ± SD with symbols = individual animals; shapes distinguish biological replicates (B, D). *, p ≤ 0.05 **, p ≤ 0.01; ***, p ≤ 0.001; ****, p ≤ 0.0001, Fisher’s Exact Test (A, C) or one-way ANOVA with Tukey test (B, D).

**Author response image 2. sa2fig2:** Pharynx tissue loss selectively increases mitotically active pharynx progenitors. (A) Proportion of *FoxA*^+^ H3P^+^ cells relative to all H3P^+^ stem cells at indicated times after pharynx or head amputation in the area outlined by dashed boxes in E. n ≥ 515 cells per experimental group from 2 independent experiments. (B) Average number of *FoxA*^+^ H3P^+^ cells quantified in the same animals and regions as A. Same data as is in Figure 3C, D of manuscript. (C) Proportion of cells double-positive for the indicated progenitor marker and H3P^+^ relative to all H3P^+^ stem cells in the area outlined by dashed boxes in E. n ≥ 472 cells per experimental group from 2 independent experiments. (D) Average number of cells double-positive for the indicated progenitor marker and H3P^+^ quantified in the same animals and regions as C. Same data as in Figure 3E of manuscript. (E) Cartoons depicting different amputation conditions. For A-D, the entire pre-pharyngeal region was analyzed over 30 z-sections, as represented by dashed boxes. Graphs represent a proportion ± 95% confidence intervals (A, C) or the mean ± SD with symbols = individual animals; shapes distinguish biological replicates (B, D). *, p ≤ 0.05 **, p ≤ 0.01; ***, p < 0.001; ****, p ≤ 0.0001, Fisher’s Exact Test (A, C) or one-way ANOVA with Tukey test (B, D)

5. Figure 2D also suffers from the inadequate quantification practices described above. Ideally, FoxA+ cells should be quantified as a percentage of the H3P+ cells observed.

We have addressed this concern as described to points 1-4 above.

6. The authors use "stem cell", "progenitor", "stem cell progenitor", and "progenitor stem cells" in a mixed and confusing way throughout the paper. For example, in lines 174-175 the authors state that "proliferation of FoxA+ stem cells precedes the increase in pharynx progenitors." This refers to FoxA+ H3P+ cells vs. FoxA+ piwi-1+ cells, but the only difference is that the former are stem cells in the act of mitosis. Is a distinction being made? Elsewhere in the paper, FoxA+ piwi+ cells are referred to as stem cells. The terminology used needs more clarity and consistency.Along these lines, what is a FoxA+ PSC (Line 168)? Are the authors suggesting that FoxA is a pluripotency marker? Or are the authors saying that tgs^-1^ has a more expansive expression pattern and is co-expressed with progenitor markers? If double FISH was performed with the progenitor markers reported (ovo, myoD, gata-4/5/6, six-1/2, and pax6), would they all overlap with tgs^-1^? These experiments need to be performed to make any claims about FoxA expression in the context of pluripotency.

We agree that our language, particularly with regard to terminology surrounding stem cells, progenitors, and organ progenitors lacked consistency. We have now modified the language in the following ways: 1) we refer to “stem cells” as *piwi-1*^+^ or H3P^+^ cells, 2) we use the term “organ-specific progenitors” to refer to cells double-positive for progenitor markers and either H3P or *piwi-1* as labeled in Figure 2 and 3, and 3) we removed all instances of the confusing terms “stem cell progenitor” and “progenitor stem cells” throughout. In addition, we removed the *tgs-1* data and any claims about pluripotent stem cells, which we agree were preliminary and inconclusive.

7. In Figure 3C, how does pharynx regeneration occur over such a long period of time after nocodazole treatment in the 1-2 day window? Does target-blind regeneration occur once this window is missed? The authors should repeat analysis of FoxA+ progenitors at later time points in this condition, and/or show rates of BrdU incorporation into the pharynx with and without nocodazole treatment in this window.

We agree that it would be valuable to know how animals eventually regenerate after perturbation of pharynx progenitors and whether non-targeted mechanisms take over to slowly regenerate this organ. However, we have struggled to devise an experiment to clearly define these parameters. Nocodazole treatment during the 1-2 day window likely results in culling of pharynx progenitors by apoptosis. Because progenitors are not undergoing mitosis at equivalently high rates in intact animals, nocodazole treatment would not perturb them similarly, making comparisons difficult. Therefore, administration of BrdU during this window could potentially label completely distinct stem cell populations.

8. The use of the inhibitor PD in Figure 4 is problematic. No data are shown to verify that phosphorylation of ERK was inhibited in these experiments. A citation of previous use is not sufficient. The effect of PD on WT uncut animals with regards to FoxA+ cells is not shown and is a necessary control. To address questions of drug specificity, the authors should corroborate their findings with a second inhibitor of ERK signaling like U0126, which has already been shown to work in planarians (Owlarn et al. 2017).

We have addressed all the concerns raised above. 1) All MEK inhibitor experiments have been repeated with UO126, with comparable results (Figure 5—figure supplement 1C-F, Figure 5—figure supplement 3B-E, Figure 6). 2) A western blot verifies that MEK inhibitor treatment for as little as 24 hours prevents ERK phosphorylation (Figure 5—figure supplement 1E). 3) We now include data showing that exposure to either inhibitor blocks regeneration in tail fragments, indicating that the pharynx regeneration defect is not due to inefficient inhibition (Figure 5—figure supplement 1F). 4) We now include quantification of *FoxA*^+^*piwi-1*^+^ in intact animals after treatment with PD and UO (Figure 5—figure supplement 2A).

9. The conclusion that ERK signaling functions in regulating differentiation but not proliferation is premature (line 249-264). Figures 4E-H should be quantified as percentages of piwi-1+ and H3P+ cells, especially since there is decreased proliferation overall with PD treatment.

In the current version of the manuscript we have made every effort to dissect the role of ERK signaling in stem cell division versus differentiation. We have now extensively tested the timeline for ERK requirement in pharynx regeneration. Based on either PD or UO exposure (Figure 5C-H, and Figure 5—figure supplement 1 and 3), our data now clearly shows that ERK activity is not required prior to the stem cell division that occurs 1-2 days after pharynx amputation. Further, even though it is true that PD and UO exposure do reduce overall stem cell division, these drug treatments do not impact division of pharynx progenitors (Figure 5H and Figure 5—figure supplement 3D,3E). Because analysis of *FoxA*^+^ H3P^+^ cells and total H3P^+^ cells are done in the same animals, one can extrapolate that the proportional data would look similar. In fact, a decrease in overall H3P would inflate the relative numbers of *FoxA*^+^ H3P^+^ cells. We have added text to clarify this in the Results section (line 426-429).

10. The use of head fragments to compare eye regeneration vs. pharynx regeneration is inappropriate. Previous studies have already shown that the absence of eyes is not required to induce ovo+ progenitor amplification (LoCascio et al. 2017). Thus, this is not a surprising result (line 343-345) and the authors are mis-citing previous observations (in the earlier work, head fragments were never described). The region where ovo+ cells was quantified in Figure 6A is not justified or explained. The yellow box is placed in a medial region where ovo+ cells do not normally reside. The authors should image within the laterally positioned ovo+ streams that have been previously described (Lapan and Reddien 2012; LoCascio et al. 2017).

We agree that this experiment did not add much to the overall findings, nor did we adequately explain our rationale for including it. It has been removed from this version.

11. Rather than using head fragments, the authors should repeat the flank resection experiments shown previously (LoCascio et al. 2017). This previous study showed that increased BrdU incorporation into the pharynx occurred following flank resection even though the pharynx was present. That result may have been 1) an artifact of increased BrdU staining due to stimulation of proliferation upon injury, 2) caused by unintentional damage to cells associated with the pharynx, or 3) a response to the loss of FoxA+ progenitor populations that surround the pharynx rather than the loss of the differentiated organ. The authors have the opportunity to revisit this published observation by quantifying the FoxA+ progenitor response during flank resection +/- pharynx. Without these data, this story is incomplete and therefore the conclusion of a targeted regeneration response is not yet convincing.

We appreciate the reviewer’s suggestion to include this important experiment. We now include quantification of *FoxA*^+^*piwi-1*^+^ cells after flank resection (Figure 2D), which proves that injuries outside of the pharynx do not trigger an increase in pharynx progenitors. Additionally, we administered EdU at different times (0hr or 24hr) after head or pharynx amputation (Figure 1E,F). We found that cells generated immediately (0hr) after amputation are incorporated broadly, reinforcing the (LoCascio et al. 2017) result that flank resection increases BrdU labeling into the uninjured pharynx. Importantly, by administering EdU 1 day after amputation, we found that cells generated at this time were channeled specifically into regenerating organs, providing evidence supporting a targeted mechanism for regeneration that selectively produces progenitors of missing tissues.

12. The negative results that proliferation and ERK signaling are dispensable for eye regeneration in Figure 6 are weak and unconvincing. The regenerated eyes appear smaller; this should be quantified (number of PRNs per eye). If the small pharynges that form in Figures 3D and 4C are considered a deleterious phenotype, why is the same standard not applied to the eye? Also, existing eye progenitors could have been sufficient for eye regeneration in these drug treatments. Furthermore, eyes did not regenerate after nocodazole treatment 50% of the time. Is it not more likely that the observations reported are dosage and timing artifacts? How has proliferation been affected? These observations do not live up to the claims made.

We have addressed potential issues with dosage and timing by standardizing the exposure times for nocodazole experiments for 2 days, increasing animal numbers, and improving the quantification. Proliferation and ERK activity are only required for eye regeneration in the context of larger amputation, but not after eye resection alone (Figure 6). Regarding the pharynx, laminin is expressed strongly in the pharynx and mouth, and weakly in the body where the pharynx attaches. Even though there is residual laminin staining 7 days after amputation with PD, UO, or nocodazole exposure (Figure 4D, 5D, Figure 5—figure supplement 1D), this is not at all a ‘pharynx’. The normal architecture, which is apparent in the DMSO controls and in the supplemental figures associated with Figures 4 and 5, is completely lost after these treatments. We have clarified this in the writing (line 290-296). Conversely, eye regeneration in drug-treated animals is undoubtedly more comparable to controls (Figure 6B).

Eye regeneration is mediated by homeostatic turnover, but whether existing eye progenitors are sufficient to regenerate these small structures after drug treatments remains unclear. We have added text in this section to address this possibility (line 450-453).

13. The authors claim that ovo+ cells are not proliferative (H3P+) even in cases where there is eye progenitor amplification (head amputation), but the data are not shown (line 321). They should be. Indeed, previous publications have never shown that ovo+ cells proliferate. This might mean that there are proliferating eye progenitors that precede expression of ovo. The authors should discuss this alternative.

We have now included images for *ovo*^+^ H3P^+^ cells in intact animals and after head amputation with or without nocodazole treatment to enrich for cells in mitosis (Figure 3—figure supplement 3). We were only able to detect dividing *ovo*^+^ cells after nocodazole treatment, but did not observe any increase after head amputation as compared to intact nocodazole treated controls. As suggested, we added discussion of the possible explanations to this section (line 265-267).

14. The authors claim that eye regeneration does not require proliferation or ERK signaling but pharynx regeneration does. This conclusion hinges on the gross observation that eyes can regenerate in the presence of nocodazole and PD (see point 12 above). These data are coarse and the interpretations are unconvincing. Instead, their model can be directly tested using BrdU pulse-chase experiments. According to the authors' model, one would predict that following pharynx amputation, the rate of incorporation of BrdU^+^ cells into the regenerating pharynx should be higher than in uninjured controls. Conversely, the rate of BrdU incorporation into the regenerating eye should remain unchanged between injured (i.e. eye-resected) and control animals (LoCascio et al. 2017). Once the authors have established the prediction above, they have the opportunity to show the effects of nocodazole, PD, and U0126 on BrdU incorporation in the regenerating eye vs pharynx following eye resection and pharynx amputation in the same animal. This way, the authors can directly test the requirement for proliferation and/or ERK signaling in both tissues.

We agree with the reviewer that this is an important experiment, and we have now addressed the different contributions of EdU^+^ cells into either the regenerating pharynx or the brain (Figure 1E,F). However, we have reserved the pharmacological perturbations for future work.

Reviewer #2:Bohr, Shiroor, and Adler investigate how stem cells respond to the loss of specific tissues in planarians. The planarian stem cell population (neoblasts) are distributed throughout the planarian body and include pluripotent stem cells and a wide range of lineage-committed progenitor cells. How this heterogenous pool of cells behave post-injury or amputation is incompletely understood. The discovery of markers to label stem cell progeny have opened the door to investigate stem cells respond to tissue loss. However, the anatomy of planarians makes it difficult to surgically remove or damage specific organs. The PI of this study developed an assay to remove the pharynx by "chemical amputation" to study the mechanisms underlying regeneration of this organ without drastically perturbing or injuring other tissues. Using this approach, this paper investigates how a well-defined population of FoxA+ progenitors respond to pharynx removal at early time-points during the regeneration. Their data suggest that stem cells are able to detect loss of the pharynx and respond by generating significantly more cells fated to become pharynx, whereas amputation of non-pharyngeal tissues does not have an obvious effect on pharynx progenitor specification dynamics. In addition, using pharmacological treatments, the authors show that cell proliferation and ERK signaling are required for the expansion of pharynx progenitors and cell differentiation. In contrast, other cell types in the planarian eye do not appear to require proliferation or ERK signaling, suggesting that stem cell responses "target blind" as suggested in a previous study, but are rather tuned to specific missing tissues.This work has the potential to make a significant contribution to the field by advancing our understanding of how the planarian heterogenous stem cell population responds to the loss of a specific organ. However, the report is preliminary as presented. It appears that the authors performed many experiments a single time. In addition, description of the methods is insufficient. Consequently, before this work can be considered for publication, the authors need to chiefly demonstrate the reproducibility of the data and robustness of the observations.1. The authors need to replicate experiments to increase the sample size for most experiments.

We have addressed this concern, as detailed in the overview above. All experiments have been repeated, increasing the sample sizes across the board. We have also included a table detailing biological and technical replicates for each relevant figure.

2. Details for imaging and quantification should be explicitly stated in the methods, and the reported cell count numbers should be normalized as appropriate for each set of experiments.

We have addressed this concern, as detailed in the overview above. Briefly, imaging and quantification methods are now described in detail within the figure legends and methods section entitled Image acquisition, quantification and statistical analysis (beginning on line 759). Regions analyzed were either of standard area or normalized to area, and for a given confocal imaging experiment, the numbers of z-sections were kept consistent.

3. Although the authors mention "lineage-tracing" experiments, they do not perform DNA analog pulse-chase experiments to analyze a temporal progression and spatial localization of stem cells to FoxA+ progenitors after pharynx removal. The authors rely on PH3-staining in conjunction with FoxA, and supplementary experiments using the pluripotent stem cell marker tgs^-1^ (which was only examined at 1 dpa). Could the authors clarify what they think FoxA+ stem cells represent? Are these self-renewing pluripotent stem cells or lineage-committed progenitors? Can the authors get some insight by scanning their images of PH3+ cells expressing FoxA visibly undergoing metaphase? Are daughter cells uniformly FoxA+ as reflected in the model? At least in one of the cells shown in the nocodazole treated controls it appears that both daughter cells express FoxA (Figure 3). I suggest showing some higher magnification images to support the interpretations/conclusions. Others have posited (e.g., Rink, Chapter 2 of Planarian Regeneration: Methods and Protocols), that not every dividing cell may be a long-term self-renewing stem cell and whether a transient amplifying cells exist or contribute to regeneration in planarians is unknown. Adler and Sánchez Alvarado (2015) discuss the role of transient states and how the transcriptional profiles change in response to regeneration. It wasn't clear to me how the authors think about these cells based on the limited number of experiments and analysis, and there are a few places where the terminology is inconsistent, especially in reference to proliferating ovo+ progenitors (P. 14). The authors need to be clear, and it might be helpful to illustrate their model in one of the early figures or to include it in the final model, which omits tgs^-1^ due to the limited number of experiments performed with this marker gene (Figure 7).

We appreciate the suggestion to clarify our language surrounding stem cells, pluripotent stem cells, and progenitors. To address the important point regarding the temporal progression of pharynx progenitors, we now include DNA analog (EdU) pulse-chase experiments to analyze the output of stem cells generated either immediately after amputation, or 1 day later when we see a specific increase in pharynx progenitors. This experiment (Figure 1E,F) shows that stem cells produced 1 day after amputation are selectively incorporated into the regenerating pharynx. As addressed in the overview above, point 4, we have made our terminology throughout the manuscript more consistent.

We agree with the reviewer regarding the complexity of the stem cell population, which includes both long-term self-renewing cells and progenitors. Because we could not make clear conclusions based on the *tgs-1* data, we removed this figure and any claims about pluripotent stem cells from this version of the manuscript. Although we do not have data to definitively indicate the potency of *FoxA*^+^ progenitors, we now include images of anaphase cells that suggest that *FoxA* can segregate into daughter cells both symmetrically and asymmetrically (Figure 3—figure supplement 1A). We have added a section addressing these issues to the discussion, because we agree that investigating these questions is important (lines 511-528).

4. The pharynx is complex and there is no data to assess what the contribution of other progenitor populations might be. I don't think or think it is unlikely that FoxA+ progenitors are solely responsible for reconstructing the pharynx. The authors should examine how other progenitor populations behave during the process of pharynx regeneration by extending the timeline of progenitor cell analysis. This would reveal if there is fluctuation in progenitor dynamics as animals regenerate the pharynx or re-scale proportions after pharynx regeneration. For example, can the authors test if they are able to detect a contribution of neural progenitors to regeneration of the pharyngeal nervous system? And if so, when during the regeneration process does it take place in the context of their study?

The reviewer is correct in noting that we still lack definitive proof of whether *FoxA*^+^ progenitors are the sole source of pharyngeal tissue or only generate a subset of critical cells. The pharynx is indeed a complex organ and we would love to know whether *FoxA* is a master regulator of them all. To address this point, we have knocked down every progenitor included in this paper and every proposed pharynx progenitor from the literature (*dd554*, *meis*, *twist*) and none of them seem to play any role in pharynx regeneration (Figure 2—figure supplement 3 and data not shown). The lack of a detectable phenotype from these markers has made us reluctant to draw conclusions from their dynamics during pharynx regeneration.

Reviewer #3:In this manuscript, Bohr et al. examine how the pluripotent stem cell system of planarians responds to organ-specific damage. If and how the differentiation of specific cell types is dynamically regulated is a conceptually fascinating problem in planarians and in general stem cell research. The authors address this problem by comparing the stem cell response between a single-organ amputation (the pharynx) versus broad tissue loss (decapitation). Their findings indicate that only the removal of the pharynx triggers the increased differentiation of pharyngeal cell types, while the loss of non-pharyngeal tissues upregulates the differentiation of progenitors of multiple organs, but not the pharynx. Further, the authors implicate temporally restricted ERK signaling as a regulatory component in the differentiation of pharyngeal cell types. These observations are also important because they contrast with the previously proposed "target blind" model (LoCascio et al., 2017) that posits the differentiation of different cell types at constant relative proportions, with the rate of stem cell divisions as global production rate regulator. In contrast, the observations by Bohr et al. provide further evidence for more flexibility and specificity within the planarian stem cell system ("target consciousness") in the sense of lineage-specific adjustments in the production rates of specific cell types.That said, the manuscript generally suffers from an overly narrow focus. Important questions remain regarding the specificity of the stem cell response to pharynx amputation and multiple experiments lack important controls (see below). Moreover, the authors have overlooked that a "target conscious" progenitor response has already been demonstrated by the selective proliferation of protonephridial marker expressing neoblasts in response to protonephridial damage by RNAi (Vu et al., 2015). As a result, the current manuscript would require substantial additional experimentation to consolidate its findings sufficiently for publication in eLife.

We appreciate the reviewer’s recognition of our work and their thoughtful criticisms. In the revised version we have added new controls, included the important reference of Vu et al., 2015, and extensively discussed the conflicting models for regeneration (referred to in our manuscript as ‘targeted’ vs ‘non-targeted’). Addressing these criticisms has resulted in a substantially more clear manuscript that adds to our understanding of the mechanisms underlying regeneration.

1. Specificity of the stem cell responseThe central premise of the paper is the selective amplification of pharyngeal progenitors in response to pharynx amputation. This the authors conclude on basis of i) an increase in the absolute number of foxA+/piwi-1+ cells in a specific area, while ii) de-capitation has no effect on the absolute number of foxA+/piwi-1+ cells in the same area. This approach is an insufficient demonstration of specificity, as the known phenomenon of wound-induced stem cell activation might also change the absolute number of specific neoblast subclasses and might do so in an injury-dependent manner.To account for this important caveat, the authors need to i) quantify RELATIVE proportions of foxA+/piwi-1+ cells out of total piwi cells (or of total H3P+ cells) and ii) they need to include other organ progenitors in the initial analysis. The latter is also critical because the pharynx is a complex organ comprising descendants of multiple lineages (e.g., muscle, neurons, epidermis) and it is not clear whether the foxA+/piwi-1+ cells indeed serve as a singular origin of all constituting lineages (as assumed by the authors), or if they only provide a subset of pharyngeal cells with a rate-limiting role in pharynx assembly (e.g., pharyngeal muscle). In face of such uncertainty, iii) the quantification of new cell incorporation into the pharynx versus other tissues via BrdU labeling would be necessary to address this caveat and to provide a global perspective on the specificity of the response independent of incompletely characterized marker genes.In addition, the following experimental design problems need to be addressed or better documented, including:• The authors provide insufficient methodological detail on progenitor quantifications, even though the entire manuscript rests on this assay. What are their criteria for scoring a piwi-1+ cell as double-positive for the often weak and noisy lineage labels? If done "by eye", was double-blind scoring used? Were all cells in a given Z-stack counted or only specific planes? If the latter, by which criteria were image planes selected for quantification? How were the specific specimens out of an experimental cohort selected for imaging/quantification? Though not necessarily a unique shortcoming of this particular study, these points simply need to be adequately addressed in order to rigorously support quantitative differences between experimental conditions (e.g., specificity).• The authors appear not to distinguish at all between technical replicates (e.g., multiple specimens within an experimental cohort) versus biological replicates (independent experimental cohorts). This is significant, because i), the use of the standard error of the mean (SEM) that the authors employ throughout is not really an appropriate measure for a single biological replicate with 3 animals – the standard deviation (SD) would seem a more appropriate measure in this context (SD). ii), the number of worms quantified for each experiment is generally low (n=3 animals in Figure 1, 3, 5, 6; n=5 animals in the rest of the figures) given the observed variability in the data (e.g., ~25-30 foxA+/piwi-1+ cells 3d after pharynx amputation in Figure 1E versus 50 cells in figure 1H). Similarly, for kinetic experiments as in Figure 1H or 2C, it is simply crucial to ensure that the error bars include the variation in response dynamics between multiple replicates due to the drift in the baseline fraction of H3P+ cells or varying staining efficiencies (e.g., different batches of animals on different days), rather than the technical variation in a single experimental cohort only. Please address these concerns by adding more specimens and a thorough description of the experimental design.

We thank the reviewer for highlighting these important caveats with our interpretation of our data. In point 2 of the introduction to this response, and in Author response figures 1 and 2, we have included a side-by-side comparison of absolute vs relative quantification and show that for key figures in our manuscript, the two strategies yield very similar results. In addition, because we are comparing different wounding paradigms that induce general wound responses, our experiments are internally controlled for wound-induced proliferation.

The reviewer is correct in noting that we still lack definitive proof of whether *FoxA*^+^ progenitors are the sole source of pharyngeal tissue or only generate a subset of critical cells (also addressed in reviewer 2’s major concern 4 above). To address this point, we have knocked down every progenitor used in this paper and every proposed pharynx progenitor from the literature (*dd554*, *meis*, *twist*). None of them cause defects in pharynx regeneration (Figure 2—figure supplement 3 and data not shown). In the current version of the manuscript, we also include pulse-chase labeling with EdU to demonstrate that stem cells are indeed selectively incorporated into the regenerating pharynx, depending on when they proliferate relative to tissue removal (Figure 1E,F). However, we acknowledge the caveat that FoxA may only control the production of a subset of cells. Ongoing work not included here is aimed at addressing this caveat.

We have also corrected all of the flaws the reviewer has pointed out related to replicates and statistical analysis in this version as outlined in the introduction to this response, point 1. Briefly, these include: 1) clarifying and standardizing methods for imaging and quantification, 2) distinguishing biological vs technical replicates in all experiments, 3) replaced error bars representing SEM with error bars representing standard deviation, and 4) increased number of animals in every experiment.

2. Timeline of pharynx regenerationThe pharynx regeneration timeline and associated events that the authors present are insufficiently supported by experimental data. The conclusion in line 215 "that proliferation in a critical window of 1 to 2 days after pharynx amputation produces a population of progenitors that are likely essential for pharynx regeneration" rests i) on the diagnosis of a "proliferative peak of FoxA+ stem cells that occurs after pharynx amputation (Figure 2C)" (line 202). However, rather than a "proliferative peak", Figure 2C shows a broad "proliferative plateau" of FoxA+ stem cells between 6h and 3 days after amputation. Similarly, the foxA+/H3P+ quantification after pharynx amputation in Figure 4G also displays a lack of a peak of foxA+/H3P+ from 1d to 2d after amputation. Ii), the associated nocodazole experiments suffer from the fact that the authors did not quantify the impact of the drug on the abundance of foxA+/piwi-1+ cells during treatment intervals from 0-1d and 2-3d after pharynx amputation. Therefore, the authors cannot rule out that nocodazole treatment might have similar effects on the abundance of foxA+/piwi-1+ cells throughout the 1-3 d post-amputation time interval, with the more severe organ-level phenotype of the 1-2d treatment window being caused by some other effect of the drug (e.g., on the differentiation of another rate-limiting cell type for pharynx regeneration or, conceivably, inhibition of priming neuronal activity ). Similar concern apply regarding the statement in line 242, "… a window 1-2 days after amputation in which activation of ERK signaling is important for pharynx regeneration." Here, i) the quantification of the end-stage phenotype of drug treatment during the 1-2d time interval (regain of feeding ability) is missing. ii) Similarly, the examination of the consequences of PD treatment on foxA+ expression in piwi-1+ cells in panel 4D-H employs drug soaking for 3 days, yet the corresponding end-stage phenotype of 3-day drug treatment is not shown. iii) the implications of ERK in pharynx regeneration are tentative. Even though the PD compound is initially correctly introduced as "MEK inhibitor", the authors subsequently switch to the factually wrong "ERK inhibitor" designation (e.g., line 358). Further, additional experimental evidence for the assumed Erk inhibition as the cause of the observed phenotypes would be desirable to rigorously support the conclusion.These caveats need to be addressed if a cell biological timeline is to remain part of this manuscript.

We thank the reviewer for their insightful criticisms. While we agree that the *FoxA*^+^ H3P^+^ peak may more closely resemble a shield volcano than a Matterhorn, regardless of terminology, there is an obvious and significant elevation in proliferation that occurs within two days of pharynx amputation. This conclusion has been strengthened by reviewers’ suggestions to boost sample size and alter statistical analysis methods. We now include additional characterization of *FoxA*^+^*piwi-1*^+^ pharynx progenitors after nocodazole treatment 0-1 and 2-3 days after pharynx amputation (Figure 4—figure supplement 2E), reinforcing our claims that proliferation occurring 1-2 days after amputation is critical for pharynx progenitor production. Additionally, this 1-2 day window is concurrent with the timing of selective incorporation of EdU^+^ into the regenerating pharynx, supporting our conclusion that this 1-2 day proliferative window is essential for pharynx progenitor production and subsequent regeneration.

Regarding ERK experiments, we have addressed every point raised by the reviewer by expanding our MEK inhibitor experiments to more thoroughly test when ERK is required during pharynx regeneration. This includes standardizing the length of MEK inhibitor treatments and performing many additional controls to show that multiple MEK inhibitors both prevent phosphorylation of ERK and lead to the same outcomes in regeneration. To bolster evidence supporting our timeline, we have done the following:

We now include data for the end-stage phenotype following both 1-2 day and 0-3 day PD exposure (Figure 5—figure supplement 3C), which both show a delay in regeneration. We also exposed animals to PD or the alternative MEK inhibitor U0126 for 24 hour intervals from 0-1, 1-2 or 2-3 days after pharynx amputation, and analyzed pharynx progenitor abundance. These experiments show that ERK inhibition between 1-2 days after amputation specifically impacts the increase of pharynx progenitors (Figure 5F, Figure 5—figure supplement 3B). Together, these experiments have strengthened our argument that ERK is required for pharynx progenitor production and pharynx regeneration.

We have modified our language to refer to the drug inhibitors as MEK inhibitors, but continue to refer to ERK inhibition, as application of MEK inhibitors does inhibit ERK activity (Owlarn et al. 2017) and phosphorylation, as shown in our western blot (Figure 5—figure supplement 1B).

3. Integration with the existing literatureThe authors need to better integrate their findings with the literature. First, they need to cite the findings of Vu et al., which explicitly demonstrated a specific increase in the fractional abundance of piwi-1+/protonephridial marker+ cells in response to RNAi-mediated damage to protonephridia(Vu et al., 2015). As such, this study already demonstrates the main point of Bohr et al., namely that the planarian stem cell system is capable of "target conscious" progenitor provision. In the very least, the authors should credit these results as additional evidence for their model. A further finding that they should discuss is the demonstration by LoCascio et al. (LoCascio et al., 2017) that flank region cut-outs cause a significant increase in pharynx cell incorporation over baseline, despite the absence of injury to the pharynx. How do the authors reconcile the discrepancy between these data and their own? In general, the discussion would benefit greatly from a more explicit comparison between the "target blind" model versus their data, as well as a broader perspective on the regulation of stem cell homeostasis.

We thank the reviewer for this suggestion, and apologize for this oversight of our colleagues’ work. In the revised version, we have more thoroughly described and contextualized our findings with the existing literature. We added sections in the introduction, discussion, and figures that specifically address the contrast between existing models for regeneration (referred to in our manuscript as targeted vs non-targeted).

We have also included experiments analyzing *FoxA*^+^*piwi-1*^+^ cells after flank resection (Figure 2D), but do not observe any increase. We have reconciled this finding with the existing literature by performing EdU labeling after different amputations, somewhat similar to those performed in (LoCascio et al. 2017), but at different times. These experiments reveal that a targeted mechanism channels cells towards missing tissues one day after amputation (Figure 1E,F), but that a non-targeted mechanism also contributes to regeneration, potentially if small amounts of tissues have been damaged (like eye resections). The first section of the discussion extensively describes these two models and how they may be utilized in planarian regeneration.